# Formation of $H_3{}^+$ from ethane dication induced by electron impact

Yu Zhang [1,2], Baihui Ren[1], Chuan-Lu Yang [3], Long Wei [1], Bo Wang[1], Jie Han[1], Wandong Yu[1], Yueying Qi[2], Yaming Zou[1], Li Chen[1], Enliang Wang [4✉] & Baoren Wei [1✉]

Hydrogen migration plays an important role in the chemistry of hydrocarbons which considerably influences their chemical functions. The migration of one or more hydrogen atoms occurring in hydrocarbon cations has an opportunity to produce the simplest polyatomic molecule, i.e. $H_3{}^+$. Here we present a combined experimental and theoretical study of $H_3{}^+$ formation dynamics from ethane dication. The experiment is performed by 300 eV electron impact ionization of ethane and a pronounced yield of $H_3{}^+ + C_2H_3{}^+$ coincidence channel is observed. The quantum chemistry calculations show that the $H_3{}^+$ formation channel can be opened on the ground-state potential energy surface of ethane dication via transition state and roaming mechanisms. The ab initio molecular dynamics simulation shows that the $H_3{}^+$ can be generated in a wide time range from 70 to 500 fs. Qualitatively, the trajectories of the fast dissociation follow the intrinsic reaction coordinate predicted by the conventional transition state theory. The roaming mechanism, compared to the transition state, occurs within a much longer timescale accompanied by nuclear motion of larger amplitude.

[1] Key Laboratory of Nuclear Physics and Ion-beam Application (MOE), Institute of Modern Physics, Fudan University, 200433 Shanghai, China. [2] School of Mathematics, Physics and Information Engineering, Jiaxing University, 314001 Jiaxing, China. [3] School of Physics and Optoelectronics Engineering, Ludong University, 264025 Yantai, China. [4] J. R. Macdonald Laboratory, Physics Department, Kansas State University, Manhattan, KS 66506, USA. ✉email: enliang@phys.ksu.edu; brwei@fudan.edu.cn

ntramolecular hydrogen or proton migration is a fundamental and ubiquitous process that plays a vital role in a wide range of fields of physics, chemistry, and biology[1–3]. It can be initiated rapidly within relevant molecules by various excitation methods, for example, photoabsorption[1], ion collisions[2], and electron impact[4]. As a consequence of the displacement of hydrogen atoms, significant structural rearrangements could be triggered, leading to isomerization and thus alter the chemical functions of those molecules. In particular, for the ionic organic molecules, numerous studies have been conducted to elucidate the formation and evolution of ultrafast hydrogen migration through the sophisticated momentum imaging techniques[5–17]. It has been found that the hydrogen-migration channel can dominate over direct fragmentation in molecular cations[5]. In addition to the well-known single hydrogen migration, another more complicated process, that is, the double hydrogen migration has been proven to play a more important role than generally assumed in molecular fragmentation[6]. Furthermore, in some of those hydrogen-migration studies special attention has been paid to the reaction dynamics in the course of $H_3^+$ formation[12–17].

$H_3^+$ has been brought into research focus by its unique structural and dynamical nature and astronomical significance as the most important interstellar medium[18,19]. In interstellar chemistry, it can protonate elementary atoms and molecules to form complex organic molecules, which could be essential for life in space. In the laboratory, $H_3^+$ ions can be produced by the fragmentation of various dicationic organic molecules induced by different ionizing projectiles or radiation (see ref. [20] and references therein). The formation of $H_3^+$ usually involves the hydrogen migration from one molecular site to another; therefore, the $H_3^+$ yield is relatively low for some molecules[20,21]. This is also confirmed by corresponding quantum chemistry calculations that the $H_3^+$ formation is generally energetically unfavorable[22].

To date, two routes to $H_3^+$ formation have been revealed via different theoretical frameworks. One is the minimum energy path (MEP) predicted by the transition state (TS) calculation. For example, Kraus et al.[23] have measured the kinetic energy release (KER) of the $H_3^+$ process for ethane dications produced by intense femtosecond laser fields. Due to the close connection to the potential energy surface (PES) of dication states, the KER can serve as a mechanistic probe through comparing with the reverse activation energy of the reaction path obtained by the TS calculation. The reaction paths for different hydrocarbon dications[20,22,23] seem to indicate that the $H_3^+$ formation proceeds through a TS representing a loosely bound complex in which an $H_2$ is attached to the remaining doubly charged moiety.

On the other hand, employing the time-resolved pump-probe technique and combining high-level ab initio molecular dynamics (AIMD) simulations[12–15], direct access has been provided to the $H_3^+$ formation dynamics and another route to $H_3^+$ formation has been proposed. Ekanayake et al.[12] have measured the time-scale, in the range of 100–260 fs, for the ejection of $H_3^+$ from laser-induced dications for a series of alcohol molecules, which was in good agreement with the AIMD simulations. The structural evolution of the dications showed that $H_3^+$ ions were formed through a roaming mechanism involving an $H_2$ molecule. The roaming pathway presents a long timescale and large region trajectory, which is far away from the MEP predicted from the conventional TS theory (TST)[24,25].

Despite the above breakthroughs, questions remain, for example, (i) could we separate the above two mechanisms to some extent and (ii) is there any other new mechanism leading to $H_3^+$ formation? To answer these questions, in this work we report an experimental and theoretical study on $H_3^+$ formation dynamics from ethane ($C_2H_6$) in collision with 300 eV electrons. We take the two-body coincidence measurement of the dissociation products to obtain the fragmentation pathway and corresponding KER distribution. On the theoretical side, we perform ab initio quantum chemistry calculations, including static energy calculations of PES and TS and AIMD simulation to retrieve structural and dynamical information of $H_3^+$ formation from the ground-state $C_2H_6^{2+}$ dications. The calculations show that the $H_3^+$ formation channel can be opened on the ground-state PES via more than one mechanism.

## Results and discussion

**Branching ratio and KER of $H_3^+ + C_2H_3^+$ channel.** The dissociative double ionization of $C_2H_6$ molecules gives rise to various fragmentation channels involving C–H and C–C bond cleavages (see Supplementary Note 1), which can be identified through the ion–ion coincidence time-of-flight (TOF) map, that is, a density plot of the TOF of the first ion vs. that of the second one. Figure 1 displays the coincidence TOF map only for $H_n^+$ ($n = 1$–3) cation channels resulting from the electron impact two-body fragmentation of ethane (see Supplementary Fig. 1 for many other channels and Supplementary Table 1 for the number of counts of all channels). The two-body $H_3^+$ channel is dominant among all $H_3^+$ channels, which is the theoretical focus of this work. As shown in Fig. 1, all of these narrow coincidence strips are aligned with the slope of −1, as expected from the two-body momentum conservation and equal charge state. The branching ratios for $H^+$, $H_2^+$, and $H_3^+$ channels are 1.1%, 26.0%, and 72.9%, respectively, as shown by the top bars of Fig. 1 (see Supplementary Note 2 and Supplementary Table 3 for theoretical branching ratios). As for all $H_n^+$ ions from the ethane dication, the branching ratios for $H^+$, $H_2^+$, and $H_3^+$ are 76.0%, 15.3%, and 8.7%, respectively (see Supplementary Table 2). The present $H_3^+$ yield is well matched with those from ethane dication formed by other charged particle collisions[26,27], but smaller than those in intense laser field ionization studies[28,29]. It is also found that the $H_3^+$ formation from ethane is much more efficient compared with some other hydrocarbon molecules, for example, $CH_4$ and $C_2H_4$[20].

After identifying the dissociation channels, the corresponding KER can be obtained employing the standard cold target recoil ion momentum spectroscopy (COLTRIMS) methodology and subsequently considering the momentum conservation constraint for the two coincidence ions. The KER distribution of the $H_3^+$ channel is plotted in Fig. 2a, which peaks at 4.7 eV. This peak value is larger than the previous results of ~4.0 and 4.3 eV obtained in strong laser field ionization[28,29]. This is probably caused by the drastic modification of the dication PES by the intense laser interaction, for example, suppressing the dissociation barrier towards the $H_3^+$ fragments and thus making the

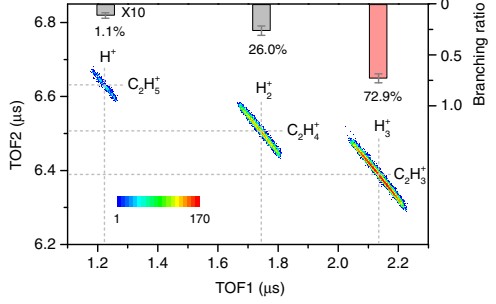

**Fig. 1 Coincidence TOF map for $H_n^+ + C_2H_{6-n}^+$ ($n = $ 1–3) channels of ethane dications.** The bars on the top show the corresponding branching ratios. The errors represent the statistical uncertainty.

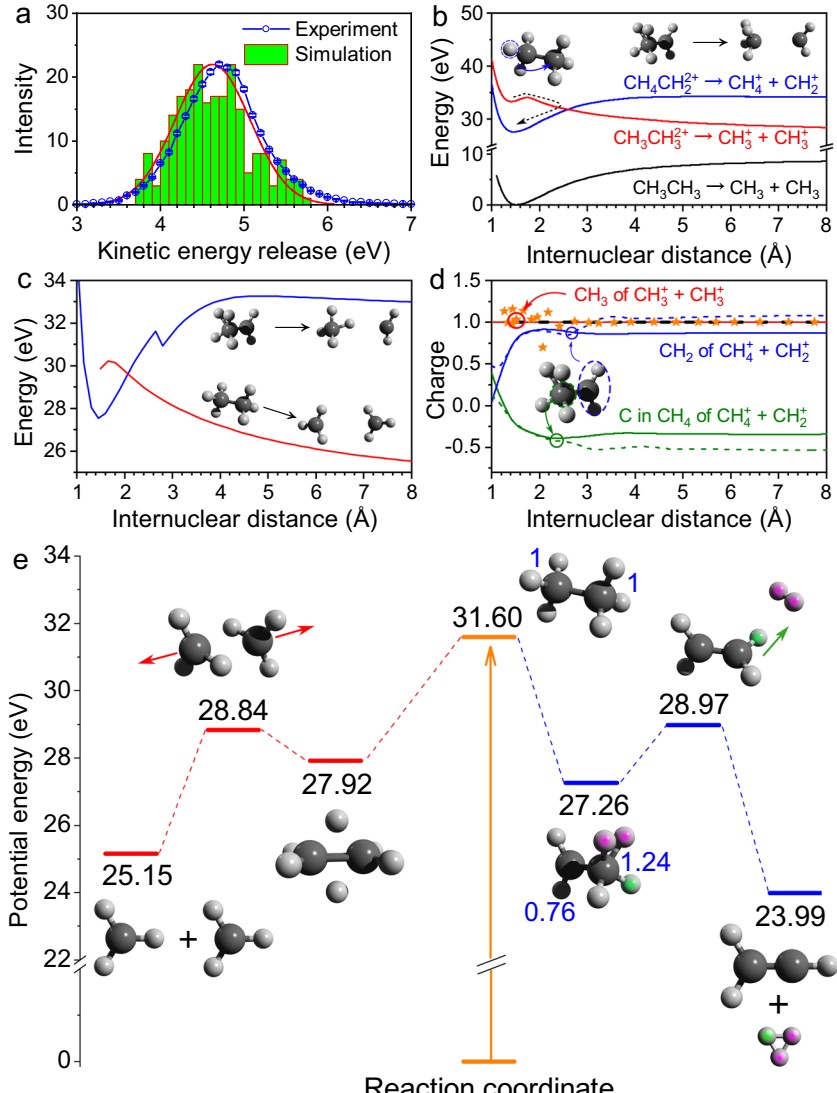

**Fig. 2 KER and reaction pathway of H$_3^+$ + C$_2$H$_3^+$ channel. a** Experimental and simulated (red curve obtained by fitting the green bars) kinetic energy release distributions for H$_3^+$ channel. **b** Potential energy curves (PECs) of C$_2$H$_6^{2+}$ dication as a function of C–C bond length by a rigid potential energy scan. The black curve is the PEC of neutral ethane ground state. The red and blue curves correspond to the PEC of dicationic ethane with dissociation limit CH$_3^+$ + CH$_3^+$ and CH$_2^+$ + CH$_4^+$, respectively. **c** PECs of C$_2$H$_6^{2+}$ dication as a function of C–C bond length by a relaxed potential energy scan. The sharp peak on the blue curve is due to the relaxation of the proton. **d** The charge distribution of CH$_3^+$ + CH$_3^+$ and CH$_2^+$ + CH$_4^+$ dissociation channel. The solid and dashed curves correspond to the rigid and relaxed potential energy scan. The orange stars come from the AIMD of CH$_3^+$ + CH$_3^+$. **e** Potential energy diagrams for the reaction path of C$_2$H$_6^{2+}$ → H$_3^+$ + C$_2$H$_3^{2+}$ (blue horizontal line) and C$_2$H$_6^{2+}$ → CH$_3^+$ + CH$_3^{2+}$ (red horizontal line). The reaction path of the transition state is confirmed by the IRC calculation and the energies are corrected by the zero-point energy. The blue numbers in **e** indicate the charge states of CH$_n$ ($n$ = 2, 3, 4). The double ionization energy discrepancies between **b**, **c**, and **e** are due to the potential energy scan without zero-point energy correction in **b**, **c** and the relaxation of degrees of freedom invalidate the vertical ionization in **c**.

original KER smaller. In Fig. 2a, the theoretical KER distribution extracted from the AIMD simulation of ground-state C$_2$H$_6^{2+}$ is superimposed with normalization at the highest point. The noticeable matching between the experimental and theoretical results indicates that the ground-state dissociation plays a prominent role in the fragmentation of C$_2$H$_6^{2+}$ dications. The crucial role of the ground-state dissociation has been also revealed in the fragmentation of other molecular dications[4,14]. Therefore, it is reasonable in the following we consider the molecular dissociation only on the ground-state PES of ethane dications. The excited states can dissociate along the adiabatic PES or decay to the ground state via internal conversion and thereby accelerate the dissociation due to the increased internal energy.

**TS calculation and AIMD simulation.** To gain more insight into the H$_3^+$ formation dynamics, we carry out quantum chemistry calculations for the dissociation of ground-state C$_2$H$_6^{2+}$. The theoretical analysis shows that the H$_3^+$ emission channel starts from a stabilization process that is similar to the case of ethanol[4]. Figure 2b shows the PESs of neutral and dicationic ethane as a function of the C–C bond length by a rigid PES scanning where all of the degrees of freedom are frozen. To confirm the dissociative limit, the charge states (Mulliken charges) localized on each group are shown as a function of bond length in Fig. 2d. The vertical double ionization from neutral ethane to the dication ([CH$_3$-CH$_3$]$^{2+}$) results in an equal charge separation of CH$_3^+$-CH$_3^+$. The charge distributions as a function of the distance of

channel $CH_3^+$-$CH_3^+$ from the rigid and relaxed scan ($C_1$ symmetry for each of the scan step) are shown in Fig. 2d by the solid-red and dashed-dark curves. Both of these two models show equal charge separation of the channel $[CH_3\text{-}CH_3]^{2+} \rightarrow CH_3^+ + CH_3^+$. Furthermore, to include the molecular vibration during the dissociation, we analyze one trajectory of $CH_3^+ + CH_3^+$ channel from the AIMD. The molecular geometries of each step are extracted and used to perform Mulliken charge analysis. The results also show an almost equal charge separation, as shown by the orange stars in Fig. 2d. The PES of $[CH_3\text{-}CH_3]^{2+}$ with the dissociation limit $CH_3^+ + CH_3^+$ is shown by the red curve in Fig. 2b. Due to such an almost repulsive PES (exhibiting a small barrier), the ethane dication can dissociate via C–C bond cleavage in principle. Before the C–C bond cleavage, an ultrafast hydrogen migration can stabilize the repulsive PES, which transforms the structure of $[CH_3\text{-}CH_3]^{2+}$ to $[CH_2\text{-}CH_4]^{2+}$ as shown by the blue curve in Fig. 2b. The experimental coincidence between $CH_3^+$ and $CH_3^+$, as well as $CH_2^+$ and $CH_4^+$, and their branching ratios (see Supplementary Fig. 1 and Supplementary Table I) confirm these two channels and the possible hydrogen migration. The following TS calculation shows that the structure of $[CH_2\text{-}CH_4]^{2+}$ plays an important role in forming $H_3^+$.

As the rigid PES scan froze all of the degrees of freedom, the obtained potential energy may deviate from the actual case. For example, the PES of $[CH_2\text{-}CH_4]^{2+} \rightarrow CH_2^+ + CH_4^+$, which is expected to be a Coulombic repulsion at long distance, shows an attraction potential up to 6 Å as shown by the blue curve in Fig. 2b. This is due to that rigid PES scan freeze all of the degrees of freedom resulting in fixed geometry of each moiety and relative orientation between them, for example, the three hydrogens are always located at one side of the carbon in $CH_4^+$ whose carbon is always facing the $CH_2^+$ as shown by the sketch on the top of Fig. 2b. There is a significant dipole moment in $CH_4^+$, which results in a dipole attractive force between $CH_4^+$ and $CH_2^+$. As shown in Fig. 2d, the charge on the carbon of $CH_4^+$ moiety is a negative value at a large distance, while the three atomic charges in $CH_2^+$ are all positive. To account for the aforementioned issues, the relaxed PES scan is performed. As shown in Fig. 2c, the blue curve shows the restrictive-relaxed PES scan of $[CH_2\text{-}CH_4]^{2+} \rightarrow CH_2^+ + CH_4^+$ where the bond lengths within $CH_4^+$ moiety are fixed, while the others degrees of freedom are relaxed. In this case, a Coulombic repulsion behavior is found at a distance >4.6 Å.

Precise energy is expected by an optimization process. Starting from the maximum of the red curve in Fig. 2b, we perform TS optimization to find the first-order saddle point on the PES and end up with the resulting TS at 28.84 eV in Fig. 2e. Finally, the reaction path is confirmed by the intrinsic reaction coordinate (IRC) and the energies are corrected by the zero-point energy. The reaction path analysis of $[CH_3\text{-}CH_3]^{2+} \rightarrow CH_3^+ + CH_3^+$ shows that after a structural rearrangement into the diborane-like double-bridged structure $H_2CH_2CH_2$ ($D_{2h}$) at 27.92 eV as shown in Fig. 2e, the C–C bond breakage encounters with the TS. As for $H_3^+$ formation, as shown in the right side of Fig. 2e, an ultrafast stabilization process is required in advance, which involves a hydrogen migration from one carbon site to another. The hydrogen migration leads to a stable dication corresponding to the local minimum on the PES of $[CH_2\text{-}CH_4]^{2+}$, as shown by the blue curve in Fig. 2b. The IRC shows that starting from the equilibrium geometry at 27.26 eV of $[CH_2\text{-}CH_4]^{2+}$, one neutral $H_2$ moiety is formed and then is emitted directly from the parent dication accompanied by the C–C bond rotation. After passing the TS at 28.97 eV, whose vibration mode corresponds to a proton transfer from carbon to $H_2$, the $H_3^+$ will finally be formed. The structure of the TS represents a loosely bound complex in a rough form of $H_2CCH_2^{2+}\cdots H_2$, that is, a quasi-neutral $H_2$ moiety attached to an ethylene "dication." The $H_2$ moiety is ~1.8 Å away from the nearest H atom and carries a total charge of $+0.20e$. It has a bond length of 0.77 Å, which is close to the bond length of the neutral hydrogen molecule, that is, 0.74 Å. In the following, the carbon from which the $H_2$ or other radical is emitted is defined as $C_\alpha$, while the other one is $C_\beta$.

The above IRC calculation of TS provides only an MEP for the molecular decomposition, while the AIMD simulation can shed more light on the fragmentation dynamics. We plot the center-of-mass distance between $H_3^+$ and $CH_2CH^+$ as a function of simulation time in Fig. 3. Qualitatively, the trajectories that follow (or very close to) the MEP are plotted in Fig. 3a, while the ones with $H_2$ undergoing a wide-range roaming are plotted in Fig. 3b, respectively. The results show that the MEP presents a faster dissociation than that of the roaming process. In this work, we define the dissociation time (marked by the red arrows in Fig. 3) as similar to the recent work of Livshits et al.[15], after which the center-of-mass distance is monotonically increasing. The typical dissociation time of MEP is <120 fs, while the roaming process dissociates between 100 and 500 fs. One typical trajectory

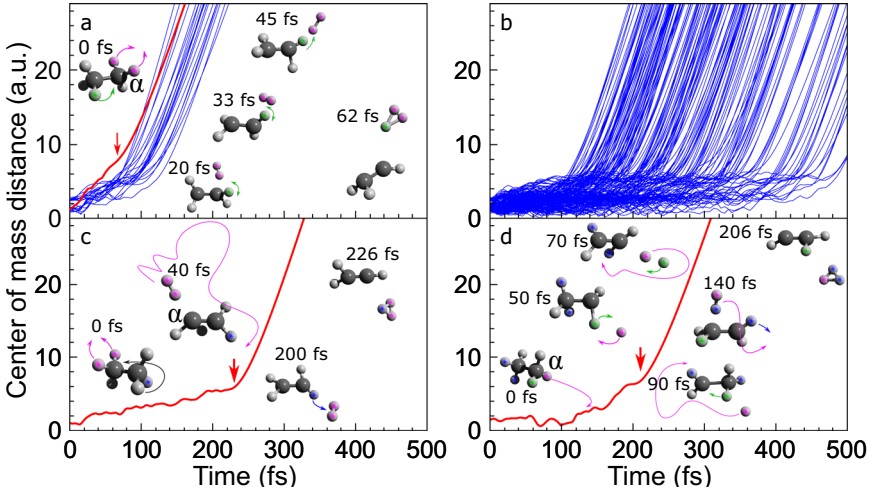

**Fig. 3 Simulated trajectories.** Center-of-mass distance between $H_3^+$ and $CH_2CH^+$ as a function of simulation time for the $H_3^+$ formation channel **a** following reaction path of the transition state and **b** from the roaming mechanism. Typical trajectories correspond to **c** $H_2$ and **d** H roaming processes. The α carbon is marked.

corresponding to the red curve in Fig. 3a is shown (see Supplementary Movie 1). Hydrogen migration from $C_\beta$ to $C_\alpha$ simultaneously with the formation of $H_2$ moiety is initiated once the dication is formed. At ~20 fs the hydrogen migration and $H_2$ formation are finished followed by vibrations between them to reach the TS at ~45 fs. Finally, an ultrafast proton absorption by the $H_2$ moiety at the $C_\alpha$ site finishes the $H_3^+$ production within 20 fs. We can see that the trajectory described above is similar to that predicted by the IRC calculation of TS shown on the right side of Fig. 2e.

As for the roaming mechanism, a long lifetime neutral radical $H_2$ is involved before $H_3^+$ emission. For the roaming process, the kinetic energies of the two products before dissociation are insufficient to overcome the nearest saddle point, that is, the TS on the PES. To dissociate the molecule, the roaming moiety may explore large regions of the PES and bypass the saddle points entirely. Besides, there may be an internal energy redistribution between the nuclei to obtain sufficient kinetic energies for the two products to overcome the potential energy barrier on the PES. To proceed with the internal energy redistribution and dynamically passing over the barrier, a longer lifetime intermediate state than the TS and wide-range movement is needed and thereby might involve roaming chemistry.

One trajectory corresponding to $H_2$ roaming is shown in Fig. 3c (see Supplementary Movie 2). The very beginning of the evolution involves an H migration from $C_\beta$ to $C_\alpha$ simultaneously with a neutral $H_2$ emission from $C_\alpha$. Then, the emitted $H_2$ starts to roam within a wide range, which lasts >100 fs. The roaming track is marked by the purple curve in Fig. 3c. At the end of the roaming, the neutral $H_2$ abstracts a proton from $C_\beta$ via an ultrafast process leading to the $H_3^+$ emission. Among all of the $H_3^+$ trajectories, we found that one particular trajectory involves the H roaming. As shown in Fig. 3d, one neutral hydrogen atom from $C_\alpha$ is ejected at the very beginning and then begins roaming (see Supplementary Movie 3). During the roaming, another H atom from $C_\alpha$ may be absorbed by the roaming one and a transient hydrogen molecule is thus formed, meanwhile an H atom migrates from $C_\beta$ to $C_\alpha$, as shown by the geometries at 50 and 70 fs in Fig. 3d. The single hydrogen roaming cannot directly result in $H_3^+$ emission. Such a process must be quenched, as shown by the geometry at 90 fs in Fig. 3d before forming a roaming neutral $H_2$. After that, the dynamics are similar to the direct $H_2$ roaming as shown in Fig. 3c.

The aforementioned mechanisms show discrepancies between IRC path of TS and roaming chemistry. Once the dication is created, the former one is activated directly where the $H_2$ radical does not move from one carbon site to another, and finally, the proton is abstracted from the parent carbon of the neutral $H_2$. The main feature of the roaming mechanism lies in, on the one hand, a rather long lifetime intermediate state that is involved, and on the other hand, there are always oscillations between neutral $H_2$ and the remaining ions. The wide-roaming range of the $H_2$ results in a free-position of the final proton abstraction, that is, either $C_\alpha$ or $C_\beta$ can be the proton donor. The direct information that can be compared is the dissociation times of these two mechanisms. Besides, the corresponding KERs seem to show some discrepancies (see Supplementary Note 3). As shown in Supplementary Fig. 2, the simulated KER of TS mechanism has the highest intensity at 4.5 eV, which seems to result in a sharp distribution, while the KER of the roaming process results in a flat distribution ~4.5 eV.

In conclusion, the $H_3^+$ formation dynamics in ethane dication is studied by combining an electron impact experiment and ab initio quantum chemical calculations. A pronounced production of $H_3^+$ ions is experimentally observed and the KER of the $H_3^+$ channel is measured. The PES calculations show that this channel can be activated on the electronic ground state of the dication,

which is confirmed by the fairly good agreement between experimental and simulated KER distributions. The TS analysis shows that $H_3^+$ is produced by a two-step evolution. In the first step, one H atom transfers from one carbon to another, which stabilizes the quasi-repulsive PES of ethane dication. Then, one neutral $H_2$ is emitted to form a transition state corresponding to a vibration mode of proton transfer from carbon to $H_2$. The forward intrinsic reaction of this transition state can produce the $H_3^+$. The dynamical analysis adopting the AIMD simulation shows that, in addition to the MEP predicted by the TST, another dominant mechanism is uncovered, that is, the roaming-induced isomerization. Comparing with the MEP, the roaming process takes a much longer time, exhibits larger amplitude motion, and requires the displacement of more H atoms. Discrepancies in dissociation time and KER of the TS and roaming mechanisms indicate that, hopefully, the two mechanisms might be separated using an ultrafast pump-probe experiment. We hope that this work would help clarify the $H_3^+$ formation mechanism in ionic molecules and provide insight into the hydrogen migration, roaming chemistry, and molecular fragmentation dynamics.

## Methods

**Experimental**. The experiment was performed by a COLTRIMS setup at Fudan University in Shanghai. Details of the setup have been described elsewhere[30]. Briefly, a pulsed electron beam with a incident energy of 300 eV perpendicularly crossed a supersonic jet of ethane in a high vacuum interaction chamber. After the interaction, the ionized ethane molecule could give rise to various dissociation channels including ionic or neutral fragments. The ionic fragments were then extracted by an electrostatic field of 60 V/cm, flew along the axis of a TOF tube, and were finally detected in coincidence by a position-sensitive detector. Three-dimensional momentum vectors of the ions were reconstructed by the detected TOF and position.

**Theoretical**. To investigate the fragmentation dynamics of ethane dication, ab initio calculations were performed for the PES and TS to analyze the pathways as well as AIMD simulation for the molecular dynamics. The calculations were carried out with the Gaussian16 program[31]. The PES, TS, and corresponding IRC were obtained by the unrestricted density functional theory (DFT) calculations using the ωB97XD functional with a dispersion correction term with the aug-cc-pVTZ basis set. All of the energy levels of the stationary points were corrected by zero-point energies. The AIMD simulation was performed under the extended Lagrangian MD scheme adopting the so-called atom-centered density matrix propagation method[32–34] using the DFT method at the B3LYP/cc-pVDZ level. The simulations were performed for the dicationic ground state resulting from the vertical ionization of the neutral ground state of ethane. Following previous studies[4,14], which have proven the importance of the ground-state dissociation, in the present study we simulated the dynamics of $H_3^+$ formation only on the ground-state PES of ethane dication. The fictitious electronic mass is 0.1 amu and the simulation time step is 0.5 fs. The initial conditions, that is, geometries and velocities of every atom, of neutral ethane were sampled by a ground-state quantum harmonic oscillator, which simulates the zero-point vibration of the molecule. In the first step, the frequencies of the neutral ethane are calculated by the Gaussian package. Then, the output file including the normal modes is transferred to the Newton-X package[35,36] to perform the initial condition sampling. A total of 1000 trajectories were computed and 230 of them end up with $H_3^+$ formation.

## Data availability

The experimental data underlying Figs. 1 and 2a are available in the figshare repository, https://doi.org/10.6084/m9.figshare.13079726.v1. The data that support the findings of this study are available from the corresponding author upon reasonable request.

## Code availability

Modified code files are available from the corresponding author upon reasonable request.

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

## Acknowledgements

This work is supported by the National Key Research and Development Program of China under Grant No. 2017YFA0402300 and the National Natural Science Foundation of China under Grant Nos. U1832201 and 11674067.

## Author contributions

Y.Z. experimented and analyzed the data. B.R., L.W., Bo Wang, and J.H. participated in the experiment. E.W. and C.-L.Y. performed the quantum chemistry calculations. Y.Z. and E.W. prepared the manuscript. W.Y., Y.Y.Q., Yaming Zou, and L.C. contributed to the interpretation of the data. Baoren Wei conceived and supervised this project. All authors participated in the discussion and review of the manuscript.

## Competing interests

The authors declare no competing interests.
