## [Peer Review File · Communications Chemistry]

Reviewers' comments:

Reviewer #1 (Remarks to the Author):

This is a nice paper demonstrating roaming dynamics in the production of H₃⁺ from the dissociation of the ethane dication. I am please to recommend publication; however, I have a couple of comments that I would like the authors to consider.

Fig.2(a) shows very nice agreement between theory and AIMD simulations of the H₃⁺ KER distribution. Could they comment on differences, if any, in the KER for the roaming vs the IRC TS pathways. Similarly is the internal energy distribution of the H₃⁺ the same or different for these pathways. I do understand that the TS pathway is "minor" and so might make a small contribution to the KER, but deconstructing it according to pathway might be very instructive.

"migrant" should be "migrate"

Reviewer #2 (Remarks to the Author):

The manuscript presents experimental and theoretical evidence regarding the formation of H₃⁺ from ethane following electron impact double ionization.

The experimental evidence is very clear, thanks to the isolation of two-body coincident dissociation events using COLTRIMS. The results in Fig. 1, are very clean. However, having removed all other coincidences, and not presenting a full mass spectrum, it would seem that 300 eV electron impact creates only three clean coincidences. All this information is missing.

- Of particular relevance to this study is the coincidence between CH₄⁺ and CH₂⁺ which would be suspected to be a major pathway for hydrogen migration.
- One also wonders about the C₂H₂⁺ and H₃⁺, and C₂H⁺ and H₃⁺ pathways, resulting from loss of one and two neutral hydrogen atoms.
- If all of these pathways are insignificantly small, then they should be mentioned, and the data should be provided as supplementary information.

Given that the dynamics being probed involve dynamics that likely occur at energies between 30-40 eV, did the authors consider a different electron impact energy?

- It seems 300 eV is excessive and the same should be observed following 70-300 eV energies, unless other states are involved. In that case, one wonders the relevance of this work to studies where the dication state is reached by laser excitation.

A great deal of the information provided about the production of H₃⁺ in this work derives from theory. This elicits the following questions:

- The authors should specify whether the DFT calculations used in the AIMD simulations were restricted or unrestricted. If restricted, the authors should discuss the fact that restricted DFT will disallow radical reactions.
- Furthermore, it is important how they prepared the initial state specifically. It seems trajectories were initiated from the ground state of the cation, presumably following vertical excitation from the neutral ground state. If this is the case, the authors should justify why initiating a process near 35 eV (Fig. 2b) or 31.39 (Fig. 2c) is relevant following 300 eV electron impact. Why the discrepancy between Figs. 2b and 2c? Could the observed results come from this an many other potential energy surfaces?
- In Fig. 2, the blue and red energy levels are not explained in the caption.
- Finally, it would be instructive to have a statistical analysis of the trajectories in terms of percent H migration vs H₂ roaming.

The manuscript may be quite relevant to the understanding of high-energy chemical reactions and

the prevalence of roaming. However, the authors should address the concerns raised here. The behavior of small hydrocarbons following high energy excitation via photons, ions, or electrons is evolving quickly. The authors should review if anything new has been published since submission.

Reviewer #3 (Remarks to the Author):

The authors describe a combined experimental and theoretical study of the ethane dication dynamics.

In the experimental part the authors provide information about the branching ratios of $H_n^+ + X^+$ products, as well as the KER distribution in the H_3^+ formation.

In the theoretical part, the authors focus on the H_3^+ formation dynamics and perform stationary TS point calculations as well as AIMD simulations. It is demonstrated that for ethane the roaming H_2 dynamics that result in H_3^+ formation are accompanied by transfer of one of the H nuclei from one carbon to the other.

The simulated mechanism shown in the manuscript is of interest. Especially as it is significantly different from previously described mechanism in the MeOH dication, in which H_2 roaming is initiated from the CH_3 part and is in competition with proton transfer. Therefore the study provides new insight and in principle is publishable.

However, I have major concerns that should be addressed:

1. The AIMD simulations are carried out on relatively lower level single-reference calculation method, compared to previous works carried out using multi-reference approaches such as CASSCF and CASPT2 methods. Furthermore, viewing the "blue" curve (fig 2b) behavior at long distances it is clear that the calculation fails to describe the ground state which asymptotic behavior in the $CH_4^+ + CH_2^+$ limit should exhibit the long range Coulombic repulsion. This apparently erroneous behavior is most likely caused due to a multireference character of the actual ground state of the dissociated system, which can not be correctly described by the DFT based methods applied in this study. Thus a highly excited state with unbalanced charge distribution is reached by DFT.

2. Nevertheless, the KER in H_3^+ formation is reasonably reproduced, indicating that it is quite possible that the important H_3^+ forming dynamics are not very significantly affected and may provide some insight although performed using a single-reference method. To further support this assumption it will be important to compare not only the KER in the H_3^+ channel, but also the reported experimental relative rates of H_3^+ , H_2^+ and H^+ formation to the simulated branching ratios. Similarly, as it was strongly emphasized in the authors discussion, it will be important and valuable to report the ratio of C=C bond breaking vs H_n^+ forming channels (and compare with exp if possible). Particular care should be taken, making sure of the fragment charge, as erroneous potentials can also result in a fictitious unbalanced charge distribution between the separated fragments.

3. In all the movies, the formation of a neutral H_2 seems to be made possible after (or in unison with) the migration of one of the hydrogens (or is it a proton ?) to form a CH_2-CH_4 structure. This point is new and interesting, as it is strikingly different from the roaming H_2 dynamics studied in methanol, in which proton migration and roaming H_2 mechanisms compete.

To convince that indeed (as proposed by the authors) the faster trajectories are going through the TS geometry, while the longer trajectories do not, it would be necessary to show some geometrical parameter as a function of time (e.g. the C=C angle rotation) that would clearly distinguish a TS mechanism from the others that go via a different path. I have to say that from the presented figures and movies I do not see a clear mechanistic distinction between short and long trajectories. And it is not convincing that the 100 fs is not an arbitrary choice, separating a single distribution into two parts. Did the authors observe a bimodal dissociation times distribution? Can these mechanisms be distinguished by their KER ?

4. Also in their introduction the authors create a contrast between two mechanisms – a transition state (TS) mechanism (e.g. reference 20) supported by KER measurements and a roaming mechanism supported by pump probe studies (e.g. refs 10-13). However, these two theoretical approaches describe practically identical dynamics:

The TS state calculated in ref 20 is actually on the path of the roaming mechanism, where a neutral H₂ is separated from the C₂H₄²⁺ dication. Thus, the roaming mechanism does proceed on the minimal energy path, in contrast to what is suggested by the authors. In fact, there's no difference between the mechanism proposed by ref 20 and the other references. The only difference is in the type of calculation performed to compare to the experimental data – an AIMD versus a TST modeling to obtain the KER.

Thus, suggesting that the literature shows two different pathways for forming H₃⁺ can mislead the readers and should be avoided.

5. Similarly, although this point is clearly seen in the roaming H movie, it would be much more convincing if the authors could present a quantitative measure that can clearly distinguish between the roaming H trajectories and the others. Thus proving that there's more than one distinct mechanism resulting in a broad dissociation time distribution.

Furthermore,

- in Fig 1: It's surprising that 300eV electrons do not result also in 3 body dissociation with H⁺ + C₂H_n⁺ with n=3 or 4. ? these are somehow missing from the coincidence map. This question relates also to the comparison of the relative probabilities of H₃⁺ / H₂⁺ and H⁺ formation with other studies e.g. ref 24 on line 97 (pg.3). As H₃⁺ formation can proceed on the ground state and 3-body fragmentation is more likely on the excited states, the H₃⁺ fraction in 2 body fragmentation can be expected to be higher than in all possible Coulomb explosion products that include also significant 3 body fragmentation where one of the fragments is an undetected neutral – see for example the different 2 and 3 body channels shown in ref 12 .

- On line 93, the end of the first paragraph of results, the authors write that the "coincidence stripes" are aligned with a -1 slope. This is must be a typo, since momentum conservation would dictate different recoil to the light and to the heavy fragment, resulting in a slope that depends on the fragment mass ratio. Inspection of fig 1 shows that indeed it is not a -1 slope. Maybe the authors intended to emphasize the anti-correlated negative slopes. Please correct the text.

- On line 105: The 4.7eV peak KER is stated to be larger than intense field studies – by how much ? is it significantly different ? what does it mean?

- In Fig 2b, the calculations made to produce the "red" and "blue" curves are not clearly defined. I assume that "red" means ground state with extension of the C-C bond from the FC geometry of the neutral CH₃-CH₃ geometry – this should be clearly explained. Were the other coordinates frozen ? or optimized ?

- The authors describe the dynamics on the "red" curve as having "high opportunity to dissociate via C-C bond 117 cleavage, due to the almost repulsive PES as a function of C-C bond length". However, having zoomed in on the figure, it seems to me that once the doubly ionized system is formed at the FC geometry, there's a significant few eV barrier for dissociation of the CC bond. In ref 12, a similar barrier is calculated for CH₃OH breakup and CC bond dissociation is attributed to dynamics on higher lying states. It seems that ethane might be similar.

- In contrast to the "red" curve, Fig2c shows a different initial energy at the neutral CC distance which is slightly lower than the "red" curve. I only guess it from the 31.39eV value in fig 2c, which is not discussed in the text. This is very confusing: Please make sure that all calculated results are explained to avoid confusing the readers. Looking at fig2c, it is now clear why the authors describe the CC bond dissociation as having "high opportunity" – although one still needs to explain the path, which is apparently not the direct dissociation presumably shown by the "red" curve and involves further specific structural rearrangement. (It is not clear from the text if this mechanism observed also in the AIMD ?)

- Fig 2c is most confusing as two very different reaction coordinates are superimposed. I strongly suggest separating the two paths – e.g. by showing one reaction coordinate in the positive X-axis direction and the other in the negative X-axis direction with respect to the initial FC geometry.

Horizontal dashed lines can help to visually compare the minima and maxima of the two reaction pathways without superimposing them.

- It is not clear how the dissociation time is defined in the discussion of fig 3 – From the figure, it appears that some of the trajectories in figure 3a (assigned with <100fs H₃⁺ formation) actually seem to form H₃⁺ at longer times (some above 150fs). Please explain how and justify the way “dissociation time” is defined.
- Actually there should be two times that should be clearly identified by AIMD analysis – the separation time of the neutral H₂ and the proton capture which is followed by the rapid dissociation. Both appear as elongation of the mean distance of the three hydrogens that from the carbon, however, H₃⁺ formation can be clearly identified by looking at the fragments relative velocity which monotonically increases after H₃⁺ is formed due to the strong Coulombic repulsion. (Similarly there should be a way to characterize the times associated with the exciting observation of roaming H dynamics)
- To continue my comment regarding the assignment of different mechanisms, it is not clear (at least from the supplied distance vs time figures 3a,b) why 100 fs was chosen – are there two clearly distinguishable dissociation time distributions ? It does not appear so from the 3a and 3b figures, making the chosen distinction seem arbitrary.
- It is not clear from figure 2c – is the transferred H is a proton or is it neutral ? Furthermore, what is the initial charge distribution at the FC geometry on the dication ground state ? is it balanced between the two CH₃⁺ parts ? or is it unbalanced as in the case of methanol ? The barrier shown by the “red” curve suggests that it might be unbalanced – similar to the methanol case where the barrier arises from the charge transfer – balancing it between the two fragments. In addition to the description of the geometrical changes during the AIMD simulation – one needs to provide more information about the charge distribution which is critical to the dynamics.
- Remarkably, as opposed to the proton migration previously reported for doubly ionized methanol – the authors provide evidence for neutral H migration (at least according to the “roaming neutral H” statement.) This should be emphasized and discussed.
- In the methods section – how many of the computed 1000 trajectories show H₃⁺ formation ?
- On line 224 – the TS model uses stationary point calculations; however the path is not stationary. Please correct the phrasing.

Response to referees

We would like to thank the referees for their careful reading, thoughtful suggestions, and thorough evaluation of our manuscript. We have addressed all comments and substantially revised the manuscript.

Reviewer #1 (Remarks to the Author):

This is a nice paper demonstrating roaming dynamics in the production of H₃⁺ from the dissociation of the ethane dication. I am please to recommend publication; however, I have a couple of comments that I would like the authors to consider.

Fig.2(a) shows very nice agreement between theory and AIMD simulations of the H₃⁺ KER distribution. Could they comment on differences, if any, in the KER for the roaming vs the IRC TS pathways. Similarly is the internal energy distribution of the H₃⁺ the same or different for these pathways. I do understand that the TS pathway is "minor" and so might make a small contribution to the KER, but deconstructing it according to pathway might be very instructive.

Response:

Thanks for the positive comments on our study. We have given the H₃⁺ KER distributions for different pathways (see section 3 in the supplementary information). The simulated KER of TS mechanism has the highest intensity at 4.5 eV which looks like to result in a sharp distribution, while the KER of the roaming process results in a flat distribution around 4.5 eV. Such a difference of KER might be observed employing an ultrafast pump-probe experiment.

"migrant" should be "migrate"

Response: We have corrected this mistake, thanks.

Reviewer #2 (Remarks to the Author):

The manuscript presents experimental and theoretical evidence regarding the formation of H₃⁺ from ethane following electron impact double ionization.

The experimental evidence is very clear, thanks to the isolation of two-body coincident dissociation events using COLTRIMS. The results in Fig. 1, are very clean. However, having removed all other coincidences, and not presenting a full mass spectrum, it would seem that 300 eV electron impact creates only three clean coincidences. All this information is missing.

- Of particular relevance to this study is the coincidence between CH₄⁺ and CH₂⁺ which would be suspected to be a major pathway for hydrogen migration.

- One also wonders about the C₂H₂⁺ and H₃⁺, and C₂H⁺ and H₃⁺ pathways, resulting from loss of one and two neutral hydrogen atoms.

- If all of these pathways are insignificantly small, then they should be mentioned, and the data should be provided as supplementary information.

Response: Figure 1 displays a small part of the full coincidence TOF map just for the targeted H_n⁺

($n=1-3$) cation channels, which result from two-body fragmentation of ethane dication. These three complete dissociation channels can be directly measured as no neutral fragment is involved. In addition, the branching ratios among these channels can be evaluated more accurately to make a comparison with those from other molecules (see Supplementary Table II).

We have presented the full coincidence TOF map in Supplementary Figure 1. According to this map, the number of counts for all ion-pair channels from the dissociation of ethane dication have been estimated in Supplementary Table I. From these counts, more information for the ethane dication dissociation can be obtained, e.g. the branching ratios among the H_n^+ ($n=1-3$) ions (see Supplementary Table II). Some of this information has been discussed in the main or supplemental texts.

- We have observed the $CH_2^+ + CH_4^+$ channel, which serves as an indicator of hydrogen migration around the skeletal C-C bond as compared to the $CH_3^+ + CH_3^+$ channel. The count of the $CH_2^+ + CH_4^+$ channel is $\sim 6.3\%$ of that of the $CH_3^+ + CH_3^+$ channel and $\sim 16\%$ of that of the interested channel $H_3^+ + C_2H_3^+$. In the main text, we focus our attention on the $H_3^+ + C_2H_3^+$ channel.

- In addition to the dominant $H_3^+ + C_2H_3^+$ channel, other H_3^+ channels involving neutral hydrogen atom(s) are also observed. Their counts are also estimated as listed in Supplementary Table I. These many-body channels may exhibit more complicated dynamics which are beyond the scope of this article.

Given that the dynamics being probed involve dynamics that likely occur at energies between 30-40 eV, did the authors consider a different electron impact energy?

- It seems 300 eV is excessive and the same should be observed following 70-300 eV energies, unless other states are involved. In that case, one wonders the relevance of this work to studies where the dication state is reached by laser excitation.

Response:

We agree that the involved dynamics occurs at energies about 30-40 eV. The selected impact electron energy of 300 eV is more due to technical requirements. At this energy, the ionization cross-section is large enough and the pulsed electron beam is stable for a long time experiment. Reading from Ref. J. Chem. Phys. 109, 1704 (1998) (Table I), all of the possible channels are opened at impact energy higher than 45 eV and the partial ionization cross-section doesn't change so much as a function of impact energy.

In electron-molecule collisions, the deposition energy exhibits a broad distribution and so is the excitation energy of the doubly ionized molecule. Thus the dication could be produced in the ground state or many other excited states. This is usually dissimilar to the case of molecular photoionization where the excitation energy and thus precursor state can be selected. However, a previous photoionization study (see Ref. [14]) indicates that the H_3^+ channel of molecular dication is mainly caused by the ground state dissociation. For the case of electron impact (see Ref. [4] at an impact energy of 91 eV), the projectile energy loss spectrum shows onsets at about 28 eV for the H_3^+ channel from ethanol dication dissociation, which means that this channel is initiated by the removal of two electrons from the outermost molecular orbital reaching the dicationic ground state. Therefore, in the present study we simulate the dynamics of H_3^+ formation only on the potential energy surface of the ground state dication, the KER is well reproduced.

Generally speaking, the comparable parameters in electron collision and the photoionization are

the electron energy loss (energy difference between incident and scattering projectile) and the photon energy rather than the incident electron energy and the photon energy.

A great deal of the information provided about the production of H₃⁺ in this work derives from theory. This elicits the following questions:

- The authors should specify whether the DFT calculations used in the AIMD simulations were restricted or unrestricted. If restricted, the authors should discuss the fact that restricted DFT will disallow radical reactions.

Response: The DFT calculations used in the AIMD simulations were unrestricted. We have specified this in the theoretical method section.

- Furthermore, it is important how they prepared the initial state specifically. It seems trajectories were initiated from the ground state of the cation, presumably following vertical excitation from the neutral ground state. If this is the case, the authors should justify why initiating a process near 35 eV (Fig. 2b) or 31.39 (Fig. 2c) is relevant following 300 eV electron impact. Why the discrepancy between Figs. 2b and 2c? Could the observed results come from this an many other potential energy surfaces?

Response:

The simulations were performed for the dicationic ground state resulting from the vertical ionization of the neutral ground state of ethane. The initial configurations, velocities, and geometries of the atoms of the neutral ethane, are obtained by analyzing the normal mode vibration of the harmonic oscillator which is equivalent to the Wigner distribution.

As mentioned earlier, in electron-molecule collisions the dication could be produced in the ground state or many other states. Thus the observed results could come from other potential energy surfaces of ethane dication. Generally, based on the fast decay approximation the excitation energy can be regarded as the internal energy of the ground state molecule (see Ref. [6]) in the simulation to avoid an expensive calculation of the excited state. The fairly good agreement between the simulated and experimental KER of our present work indicates the initial condition adopted is reasonable.

The energy discrepancies between Fig. 2b and 2c are due to the following reasons: (1) Fig. 2b is the electronic energy without zero-point-energy (ZPE) correction while Fig. 2c includes the ZPE correction; (2) Fig. 2b is the rigid potential energy scan where all of the degrees of freedom are fixed while in Fig. 2c all of the stationary points (except for 31.39 eV) are obtained by geometry optimization. Strictly speaking, the energies in Fig. 2c are better than those in Fig. 2b. Fig. 2b is more a physical picture analysis showing that CH₃CH₃²⁺ is more likely to transform into CH₄CH₂²⁺ which is crucial for the production of H₃⁺.

- In Fig. 2, the blue and red energy levels are not explained in the caption.

Response: These energy levels have been explained in the caption.

- Finally, it would be instructive to have a statistical analysis of the trajectories in terms of percent

H migration vs H2 roaming.

Response: The H roaming is a very special mechanism and only one trajectory is confirmed among the 230 trajectories. In the revised manuscript we clarify that we only confirm one trajectory of this mechanism.

The manuscript may be quite relevant to the understanding of high-energy chemical reactions and the prevalence of roaming. However, the authors should address the concerns raised here. The behavior of small hydrocarbons following high energy excitation via photons, ions, or electrons is evolving quickly. The authors should review if anything new has been published since submission.

Response: We agree that the dynamics of small hydrocarbons by various excitation methods are a very hot topic. We have cited pertinent works in the revised manuscript.

Reviewer #3 (Remarks to the Author):

The authors describe a combined experimental and theoretical study of the ethane dication dynamics.

In the experimental part the authors provide information about the branching ratios of $H_n^+ + X^+$ products, as well as the KER distribution in the H_3^+ formation.

In the theoretical part, the authors focus on the H_3^+ formation dynamics and perform stationary TS point calculations as well as AIMD simulations. It is demonstrated that for ethane the roaming H_2 dynamics that result in H_3^+ formation are accompanied by transfer of one of the H nuclei from one carbon to the other.

The simulated mechanism shown in the manuscript is of interest. Especially as it is significantly different from previously described mechanism in the MeOH dication, in which H_2 roaming is initiated from the CH_3 part and is in competition with proton transfer. Therefore the study provides new insight and in principle is publishable.

Response: We thank the referee for the positive comments on our manuscript.

However, I have major concerns that should be addressed:

1. The AIMD simulations are carried out on relatively lower level single-reference calculation method, compared to previous works carried out using multi-reference approaches such as CASSCF and CASPT2 methods. Furthermore, viewing the “blue” curve (fig 2b) behavior at long distances it is clear that the calculation fails to describe the ground state which asymptotic behavior in the $CH_4^+ + CH_2^+$ limit should exhibit the long range Coulombic repulsion. This apparently erroneous behavior is most likely caused due to a multireference character of the actual ground state of the dissociated system, which cannot be correctly described by the DFT based methods applied in this study. Thus a highly excited state with unbalanced charge distribution is reached by DFT.

Response:

The CASSCF and CASPT2 works of Luzon *et al.* (Ref. [14]) and Livshits *et al.* (Ref. [15])

showed a fairly good description of fragmentation dynamics of methanol. However, due to the limited computational resources, it's hard for us to adopt the multi-reference method to simulate the dynamics for thousands of trajectories within a limited time. The propagation method like ADMP (atom-centered density matrix propagation which is similar to CPMD (Car–Parrinello molecular dynamics)) performed by DFT are reasonable choices for an order of thousand or even ten thousand trajectories. Such a DFT method at the same or similar level has been widely used to study the ultrafast hydrogen migration and the formation of H_3^+ , H_3O^+ ions from organic dications produced by electron impact (Ref. [4]), multiply charged ion collision (*J. Phys. Chem. Lett.* **4**, 3903 (2013)), and photoionization (Ref. [6]). The fairly good agreement between the theoretical and experimental KERs indicates that the calculation level of our present work is not bad.

For the potential energy curve of the dication, we analyze the charge distribution of $CH_2^+-CH_4^+$ channel as a function of bond length. The dissociation limit corresponds to $1+$ and $1+$ as shown by the following figures.

The potential energy starts to go down at C-C distance larger than 10 Å. In Fig. 2b, we perform the rigid potential energy scanning where all of the degrees of freedom are frozen. The obtained results might be different from the actual situation. In this figure, we want to show within or nearby the Franck-Condon region the hydrogen migration can stabilize the dication to form $[CH_2-CH_4]^{2+}$ which is crucial to forming the following H_3^+ .

For the dissociation of $[CH_2-CH_4]^{2+}$ into $CH_2^+ + CH_4^+$, a potential energy barrier which is close to the most probable vertical transition point is encountered. This channel is not likely to result from the dissociation of ethane dication in the ground state. This might explain that the experimentally observed $CH_2^+ + CH_4^+$ channel (see Supplementary Figure 1) is not accessed in the present AIMD simulations performed on the ground state potential energy surface.

2. Nevertheless, the KER in H_3^+ formation is reasonably reproduced, indicating that it is quite possible that the important H_3^+ forming dynamics are not very significantly affected and may provide some insight although performed using a single-reference method. To further support this assumption it will be important to compare not only the KER in the H_3^+ channel, but also the reported experimental relative rates of H_3^+ , H_2^+ and H^+ formation to the simulated branching ratios. Similarly, as it was strongly emphasized in the authors discussion, it will be important and valuable to report the ratio of C=C bond breaking vs H_n^+ forming channels (and compare with exp if possible). Particular care should be taken, making sure of the fragment charge, as

erroneous potentials can also result in a fictitious unbalanced charge distribution between the separated fragments.

Response:

First, we agree that we must pay particular care to analyze the final state of H₂. Even the methods include the configuration interaction also provide a large divergence of the branching ratios of H₂ (see Table 1 in Re. [12]). The simulation results at 500 fs show that H₂ is produced in fractional charge (mostly 0.5 e). For the H₃⁺, it must be produced in 1+ state, (the neutral state of H₃ is not stable). For H⁺, the results show that it is produced in the 1+ state for ethane dication.

The counts or branching ratios of the C=C bond breaking and H_n⁺ forming channels of ethane dication have been summarized in Supplementary Table I~III. Please keep in mind that in the present study we just simulate the dissociation dynamics of ethane dication in the ground state, which has been proven crucial to the formation of H₃⁺ ions. In fact, in electron-molecule collisions the ethane dication could be populated in various states, which favor different dissociation channels. Therefore the present theoretical branching ratios extracted from the simulation of ground state do not agree well with the experimental ones.

Adopting the Mulliken population scheme, the fragment charge is obtained as the sum of the atomic charges in the corresponding fragment. Especially, we count all of the channels ending up with H₂ (might in neutral and charge states) which correspond to H₂⁺ + C₂H₄⁺ and H₂ + C₂H₄²⁺ dissociation limit.

3. In all the movies, the formation of a neutral H2 seems to be made possible after (or in unison with) the migration of one of the hydrogens (or is it a proton ?) to form a CH2-CH4 structure. This point is new and interesting, as it is strikingly different from the roaming H2 dynamics studied in methanol, in which proton migration and roaming H2 mechanisms compete.

To convince that indeed (as proposed by the authors) the faster trajectories are going through the TS geometry, while the longer trajectories do not, it would be necessary to show some geometrical parameter as a function of time (e.g. the C=C angle rotation) that would clearly distinguish a TS mechanism from the others that go via a different path. I have to say that from the presented figures and movies I do not see a clear mechanistic distinction between short and long trajectories. And it is not convincing that the 100 fs is not an arbitrary choice, separating a single distribution into two parts. Did the authors observe a bimodal dissociation time distribution? Can these mechanisms be distinguished by their KER?

Response:

The single H migration results in CH₂⁺-CH₄⁺ structure, which is more stable than CH₃⁺-CH₃⁺ and the H migration is an ultrafast process. Therefore the geometry transformation is the most probable process due to its the largest energy gradient. The experimental coincidence channel of CH₂⁺ and CH₄⁺, as shown in Supplementary Figure 1, confirms this possible geometry.

Generally speaking, if the H doesn't dissociate from its parent ion, it should be in a neutral state. Otherwise, there will be a Coulomb explosion due to the Coulomb repulsion. Therefore we want to ascribe this migration process as hydrogen migration.

There is no clear gap between TS and roaming mechanisms. Thus we agree that the choice of 100 fs is not a strict method. In the revised manuscript, we select the TS mechanism by

monitoring all of the H_3^+ trajectories. This is better than the single criterion of 100 fs. Of course, the method is qualitative rather than quantitative.

We don't observe a bimodal dissociation time distribution.

We analyze the KER distributions corresponding to TS and roaming mechanisms. As shown in Supplementary Figure 2, the KER of TS mechanism has the highest intensity at 4.5 eV which looks like to result in a sharp distribution, while the KER of the roaming process results in a flat distribution around 4.5 eV. We recommend a pump-probe experiment where there will be different KER distributions as a function of delay.

The manuscript is revised accordingly.

4. Also in their introduction the authors create a contrast between two mechanisms – a transition state (TS) mechanism (e.g. reference 20) supported by KER measurements and a roaming mechanism supported by pump probe studies (e.g. refs 10-13). However, these two theoretical approaches describe practically identical dynamics:

The TS state calculated in ref 20 is actually on the path of the roaming mechanism, where a neutral H_2 is separated from the $C_2H_4^{2+}$ dication. Thus, the roaming mechanism does proceed on the minimal energy path, in contrast to what is suggested by the authors. In fact, there's no difference between the mechanism proposed by ref 20 and the other references. The only difference is in the type of calculation performed to compare to the experimental data – an AIMD versus a TST modeling to obtain the KER.

Thus, suggesting that the literature shows two different pathways for forming H_3^+ can mislead the readers and should be avoided.

Response: We have revised the manuscript accordingly. The sentences to create a contrast between the two mechanisms are revised. On the other hand, we suggest that the dissociation times and KER distributions of these two mechanisms may be observed by a pump-probe experiment.

5. Similarly, although this point is clearly seen in the roaming H movie, it would be much more convincing if the authors could present a quantitative measure that can clearly distinguish between the roaming H trajectories and the others. Thus proving that there's more than one distinct mechanism resulting in a broad dissociation time distribution.

Response: We analyze all of the trajectories of H_3^+ channel and we only confirm one for the H roaming. It's not our purpose to separate the H roaming and H_2 roaming in this work because even we ascribe this trajectory to H roaming we also see the accompanying H_2 roaming as shown in Fig. 3d. A possible way to distinguish the H roaming is also expected by the pump-probe experiment where the large range of H roaming will result in low kinetic energy proton.

Furthermore,

• in Fig 1: It's surprising that 300eV electrons do not result also in 3 body dissociation with $H^+ + C_2H_n^+$ with $n=3$ or 4. ? these are somehow missing from the coincidence map. This question relates also to the comparison of the relative probabilities of H_3^+ / H_2^+ and H^+ formation with other studies e.g. ref 24 on line 97 (pg.3). As H_3^+ formation can proceed on the ground state and

3-body fragmentation is more likely on the excited states, the H_3^+ fraction in 2 body fragmentation can be expected to be higher than in all possible Coulomb explosion products that include also significant 3 body fragmentation where one of the fragments is an undetected neutral – see for example the different 2 and 3 body channels shown in ref 12 .

Response:

Figure 1 displays a small part of the full coincidence TOF map just for three two-body H_n^+ ($n=1-3$) cation channels. The full coincidence TOF map is presented in Supplementary Figure 1 where includes many three- or many-body dissociation channels. The number of counts for all dissociation channels has been estimated in Supplementary Table I. And thus the branching ratios among the H_n^+ ($n=1-3$) ions can be obtained (see Supplementary Table II), which has been discussed in the text.

In addition to the two-body channel $H_3^+ + C_2H_3^+$, other H_3^+ channels involving neutral hydrogen atom(s), e.g. $H_3^+ + C_2H_2^+/C_2H^+/CH^+$, are also observed. The fraction of $H_3^+ + C_2H_3^+$ is indeed much higher than those of other H_3^+ channels (see Supplementary Table I).

• On line 93, the end of the first paragraph of results, the authors write that the “coincidence stripes” are aligned with a -1 slope. This is must be a typo, since momentum conservation would dictate different recoil to the light and to the heavy fragment, resulting in a slope that depends on the fragment mass ratio. Inspection of fig 1 shows that indeed it is not a -1 slope. Maybe the authors intended to emphasize the anti-correlated negative slopes. Please correct the text.

Response: In our experiment, we only detect the ionic fragments from the molecule while the scattering projectile and ejected electrons are omitted. The final state electrons have only a small influence on the momentum conservation of the fragments. The magnitude of electron momentum is on the order of $\sim a.u.$, while that of the fragment is in the order of several tens to hundreds of a.u. Please note that the vertical and horizontal scalars for TOF are different in Fig. 1. The slope of these three narrow coincidence stripes is indeed -1 (see the figure attached below for referee’s reference). The slope of the orange line is $(6.8 - 6.2)/(1.1 - 1.7) = -1$. The H_n^+ coincidence strips aligned along the other three blue lines, which are parallel to the orange one, is indeed tilted with a slope of -1.

• On line 105: The 4.7eV peak KER is stated to be larger than intense field studies – by how much ? is it significantly different ? what does it mean?

Response: We have modified this sentence as “This peak value is larger than the previous results of about 4.0 and 4.3 eV obtained in strong laser field ionization [24,25]. This is probably caused by the drastic modification of the dication PES by the intense laser interaction, e.g., suppressing the dissociation barrier towards the H_3^+ fragments and thus making the original KER smaller.”

• In Fig 2b, the calculations made to produce the “red” and “blue” curves are not clearly defined. I assume that “red” means ground state with extension of the C-C bond from the FC geometry of the neutral CH_3-CH_3 geometry – this should be clearly explained. Were the other coordinates frozen ? or optimized ?

Response: We have defined the red and blue curves and explained them clearly in the figure caption. When extending the C-C internuclear distance, the other coordinates were frozen.

• The authors describe the dynamics on the “red” curve as having “high opportunity to dissociate via C-C bond 117 cleavage, due to the almost repulsive PES as a function of C-C bond length”. However, having zoomed in on the figure, it seems to me that once the doubly ionized system is formed at the FC geometry, there’s a significant few eV barrier for dissociation of the CC bond. In ref 12, a similar barrier is calculated for CH_3OH breakup and CC bond dissociation is attributed to dynamics on higher lying states. It seems that ethane might be similar.

• In contrast to the “red” curve, Fig2c shows a different initial energy at the neutral CC distance which is slightly lower than the “red” curve. I only guess it from the 31.39eV value in fig 2c, which is not discussed in the text. This is very confusing: Please make sure that all calculated results are explained to avoid confusing the readers. Looking at fig2c, it is now clear why the authors describe the CC bond dissociation as having “high opportunity” – although one still needs to explain the path, which is apparently not the direct dissociation presumably shown by the “red” curve and involves further specific structural rearrangement. (It is not clear from the text if this mechanism observed also in the AIMD ?)

Response:

Both of the above two comments are related to the Figs. 2b and 2c, so here we address them as a whole. Below is the reorganized discussion concerning the potential energy curves or diagrams as shown in Figs. 2b and 2c.

As shown by the reaction equations in Fig. 2b, the blue and red curves correspond to two reaction channels. This is a rigid potential energy scanning that freezes all of the degrees of freedom. Thus the energy may be different from the optimized one. This figure gives us a rough picture that, without hydrogen migration, the dication ion will dissociate into CH_3^+ and CH_3^+ along C-C bond stretching. The crossing between the red and blue curves indicates that hydrogen migration can transform the structure $[CH_3-CH_3]^{2+}$ to $[CH_2-CH_4]^{2+}$ where the latter is more stable and is the starting point of the H_3^+ emission channel as shown in Fig. 2c. The energy discrepancies

between Figs. 2b and 2c are due to the following reasons: (1) Fig. 2b is the electronic energy without zero-point-energy (ZPE) correction while Fig. 2c includes the ZPE correction; (2) Fig. 2b is the rigid potential energy scan where all of the degrees of freedom are fixed while in Fig. 2c all of the stationary points (except for 31.39 eV) are obtained by geometry optimization. Strictly speaking, the energies in Fig. 2c are better than those in Fig. 2b. Fig. 2b is more a physical picture analysis showing that $\text{CH}_3\text{CH}_3^{2+}$ is more likely to transform into $\text{CH}_4\text{CH}_2^{2+}$ which is crucial for the production of H_3^+ .

The associated texts in the manuscript have been revised to avoid misleading.

• Fig 2c is most confusing as two very different reaction coordinates are superimposed. I strongly suggest separating the two paths – e.g. by showing one reaction coordinate in the positive X-axis direction and the other in the negative X-axis direction with respect to the initial FC geometry. Horizontal dashed lines can help to visually compare the minima and maxima of the two reaction pathways without superimposing them.

Response: We have replaced Fig. 2c with a new version according to the referee's suggestion.

• It is not clear how the dissociation time is defined in the discussion of fig 3 – From the figure, it appears that some of the trajectories in figure 3a (assigned with <100fs H3+ formation) actually seem to form H3+ at longer times (some above 150fs). Please explain how and justify the way “dissociation time” is defined.

• Actually there should be two times that should be clearly identified by AIMD analysis – the separation time of the neutral H2 and the proton capture which is followed by the rapid dissociation. Both appear as elongation of the mean distance of the three hydrogens that from the carbon, however, H3+ formation can be clearly identified by looking at the fragments relative velocity which monotonically increases after H3+ is formed due to the strong Coulombic repulsion. (Similarly there should be a way to characterize the times associated with the exciting observation of roaming H dynamics)

• To continue my comment regarding the assignment of different mechanisms, it is not clear (at least from the supplied distance vs time figures 3a,b) why 100 fs was chosen – are there two clearly distinguishable dissociation time distributions ? It does not appear so from the 3a and 3b figures, making the chosen distinction seem arbitrary.

Response:

The above three comments point to the same concern, so here we address them as a whole.

We agree that the choice of 100 fs seems arbitrary. As mentioned before, in the revised manuscript we reanalyze all of the trajectories of H_3^+ channels, and the ones following (or very close to) the reaction coordinate of TS are ascribed to the TS mechanism. One more reasonable way to define the dissociation time is shown in Ref. Communications Chemistry 3, 49 (2020) where the oscillation-structure disappearing point on the relative velocity was defined as the dissociation time. Similarly, we define the dissociation time as the oscillation-structure disappearing point on the center of mass distance in the revised manuscript.

• *It is not clear from figure 2c – is the transferred H is a proton or is it neutral ? Furthermore, what is the initial charge distribution at the FC geometry on the dication ground state ? is it balanced between the two CH₃⁺ parts ? or is it unbalanced as in the case of methanol ? The barrier shown by the “red” curve suggests that it might be unbalanced – similar to the methanol case where the barrier arises from the charge transfer – balancing it between the two fragments. In addition to the description of the geometrical changes during the AIMD simulation – one needs to provide more information about the charge distribution which is critical to the dynamics.*

Response: The required charge states are shown in Fig. 2c. The charge distribution at the FC geometry on the dication ground state is balanced as a result of the symmetry of two CH₃⁺. But for the [CH₂-CH₄]²⁺, the CH₂ part carries a total charge of +0.85e and the CH₄ part carries +1.15e according to the Mulliken charge analysis at wb97xd/cc-pVTZ level. This means the transferred hydrogen atom leading to the isomerization from [CH₃-CH₃]²⁺ to [CH₂-CH₄]²⁺ carries a small fraction of one positive charge. We prefer to identify the transferred hydrogen as a neutral one.

• *Remarkably, as opposed to the proton migration previously reported for doubly ionized methanol – the authors provide evidence for neutral H migration (at least according to the “roaming neutral H” statement.) This should be emphasized and discussed.*

Response: We have modified corresponding sentences to discuss the H roaming process properly.

• *In the methods section – how many of the computed 1000 trajectories show H₃⁺ formation ?*

Response: We have added a sentence in the theoretical method section “A total of 1000 trajectories was computed and 230 of them end up with H₃⁺ formation.”.

• *On line 224 – the TS model uses stationary point calculations; however the path is not stationary. Please correct the phrasing.*

Response: We have modified this sentence.

Reviewers' comments:

Reviewer #1 (Remarks to the Author):

The authors have done a thorough revision and I believe have addressed all concerns. Clearly the roaming pathway to H₃⁺ is not a major one at the energy of the experiment; however, it is nicely supported by the trajectory simulations. I would also note that in general lifetimes are not definitive signatures of "roaming", which in fact can be quite prompt.

I do recommend publication.

Reviewer #2 (Remarks to the Author):

The authors have addressed each of the questions in detail to the best of their available information. There may still be questions that remain unanswered and will need new experiments and new calculations, but those are not needed to accept the paper for publication.

Reviewer #3 (Remarks to the Author):

The authors clearly made an effort to improve their manuscript. Nevertheless, in my opinion, at least some of the details included in response to the review, should be addressed also in the manuscript itself.

In particular, I think that the charge distribution presented in the authors response letter should be included at least in the supporting information.

I find it highly unlikely and surprising that two cations (with frozen coordinates) attract each other at 3-6Å distances - as shown by the blue curve in figure 2b. I would expect the potential to be dominated by the long range Coulombic repulsion at this long distance. Nevertheless, the authors claim in their response that the charge is almost equally distributed between a CH₂⁺ and CH₄⁺ fragments. I suggest that the authors should at least point out this surprising behaviour to the readers in the manuscript text itself and add the charge distribution they calculated to support their assignment. (to be clear, I'm attaching a copy of that figure)

One more point I would like to contribute is the fact that a system is symmetric does not necessarily result in equal charge distribution. The symmetric CH₃-CH₃ system can easily find itself in a superposition of non-symmetric states: i.e. dication-neutral + neutral-dication. Furthermore, considering the role of symmetry, it might be instructive to perform static PES calculations not only in the symmetric minimum energy geometry , but also from other regions in the FC region in which symmetry can be broken by the zero point motion.

Response to referees

We would like to thank the referees for their careful reading, thoughtful suggestions, and thorough evaluation of our revised manuscript. We have addressed all comments and revised the manuscript again. The revised sentences are highlighted in blue color.

Reviewer #1 (Remarks to the Author):

The authors have done a thorough revision and I believe have addressed all concerns. Clearly the roaming pathway to H_3^+ is not a major one at the energy of the experiment; however, it is nicely supported by the trajectory simulations. I would also note that in general lifetimes are not definitive signatures of "roaming", which in fact can be quite prompt.

I do recommend publication.

Our reply:

We thank the reviewer for the positive evaluation of our manuscript. We agree that, in general, lifetimes are not definitive signatures of "roaming", which in fact can be quite prompt. As shown by the trajectories of our simulation, a small part of the roaming mechanism emit the H_3^+ near 100 fs which is an even fast process. In the revised manuscript, we modify some sentences which are highlighted in blue color (line 28, and line 231).

Reviewer #2 (Remarks to the Author):

The authors have addressed each of the questions in detail to the best of their available information. There may still be questions that remain unanswered and will need new experiments and new calculations, but those are not needed to accept the paper for publication.

Our reply:

We thank the reviewer for the positive evaluation of our manuscript.

Reviewer #3 (Remarks to the Author):

The authors clearly made an effort to improve their manuscript.

Nevertheless, in my opinion, at least some of the details included in response to the review, should be addressed also in the manuscript itself.

In particular, I think that the charge distribution presented in the authors response letter should be included at least in the supporting information.

I find it highly unlikely and surprising that two cations (with frozen coordinates) attract each other at 3-6Å distances - as shown by the blue curve in figure 2b. I would expect the potential to be dominated by the long range Coulombic repulsion at this long distance. Nevertheless, the authors claim in their response that the charge is almost equally distributed between a CH₂⁺ and CH₄⁺ fragments. I suggest that the authors should at least point out this surprising behaviour to the readers in the manuscript text itself and add the charge distribution they calculated to support their assignment. (to be clear, I'm attaching a copy of that figure) One more point I would like to contribute is the fact that a system is symmetric does not necessarily result in equal charge distribution. The symmetric CH₃-CH₃ system can easily find itself in a superposition of non-symmetric states: i.e. dication-neutral + neutral-dication. Furthermore, considering the role of symmetry, it might be instructive to perform static PES calculations not only in the symmetric minimum energy geometry , but also from other regions in the FC region in which symmetry can be broken by the zero point motion.

Our reply:

We thank the reviewer for the informative comments and suggestions.

- 1. We revise Figure 2 in the manuscript to include the relaxed potential energy scanning and the charge distribution as a function of bond length. The corresponding manuscript is revised (line 130-140 and line 150-162).*
- 2. For the no Coulombic repulsion behavior at a large distance, first, we want to exclude the influence of the basis set. We perform the calculations using ω B97XD/aug-cc-pVTZ which includes the diffuse basis set. Generally, the missing diffuse basis set may result in an inaccurate description of the potential energy at a very large distance. The new results show similar behavior as that of ω B97XD/cc-pVTZ. Here, we ascribe the no Coulombic repulsion behavior due to the frozen of all of the degrees of freedom. In this case, the geometries of each moiety and their relative orientation are fixed which might result in a large dipole attractive force between them. We analyze the charge distribution of this CH₄⁺. At a large distance (e.g. 3-6 Å), carbon is in a negative charge state which will attract the facing CH₂⁺ moiety. We perform a relaxed PEC scan in the revised manuscript.*

The new result shows that the Coulombic repulsion behavior is obtained at distance larger than 4.6 Å. In the revised manuscript, the data are updated to the results of ω B97XD/aug-cc-pVTZ.

- 3. To break the symmetry, we perform a relaxed PES scan without symmetry restriction of the $\text{CH}_3^+ + \text{CH}_3^+$ channel. For each of the scan steps, the symmetry is C_1 and equal charge separation is also obtained. Furthermore, to include the molecular vibration during the dissociation we simulate one trajectory of $\text{CH}_3^+ + \text{CH}_3^+$ channel using ADMP (B3LYP/cc-pVDZ) method. The molecular geometries of each step are extracted and used to perform Mulliken charge analysis by ω B97XD/aug-cc-pVTZ. The results also show an almost equal charge separation for the dissociation limit. The expression of the relationship between charge and symmetry is revised. For the ground state of the dication of ethane $[\text{CH}_3\text{-CH}_3]^{2+}$, the density functional method together with the aug-cc-pVTZ basis provides a good description of the wavefunction and thereby the charge state. We believe the “dication-neutral + neutral-dication” configuration will result in no-equal charge separation. Such a configuration, however, is most likely corresponding to an excited state which is out of our current discussion.*

REVIEWERS' COMMENTS:

Reviewer #3 (Remarks to the Author):

As requested, the authors discussed the unexpected shape of the PES curve, making important improvements to the figures and text of the manuscript.

It is now suitable for publication as the limitations of the theoretical treatment are clearly emphasized.

Reviewers' comments:

Reviewer #1 (Remarks to the Author):

This is a nice paper demonstrating roaming dynamics in the production of H₃⁺ from the dissociation of the ethane dication. I am please to recommend publication; however, I have a couple of comments that I would like the authors to consider.

Fig.2(a) shows very nice agreement between theory and AIMD simulations of the H₃⁺ KER distribution. Could they comment on differences, if any, in the KER for the roaming vs the IRC TS pathways. Similarly is the internal energy distribution of the H₃⁺ the same or different for these pathways. I do understand that the TS pathway is "minor" and so might make a small contribution to the KER, but deconstructing it according to pathway might be very instructive.

"migrant" should be "migrate"

Reviewer #2 (Remarks to the Author):

The manuscript presents experimental and theoretical evidence regarding the formation of H₃⁺ from ethane following electron impact double ionization.

The experimental evidence is very clear, thanks to the isolation of two-body coincident dissociation events using COLTRIMS. The results in Fig. 1, are very clean. However, having removed all other coincidences, and not presenting a full mass spectrum, it would seem that 300 eV electron impact creates only three clean coincidences. All this information is missing.

- Of particular relevance to this study is the coincidence between CH₄⁺ and CH₂⁺ which would be suspected to be a major pathway for hydrogen migration.
- One also wonders about the C₂H₂⁺ and H₃⁺, and C₂H⁺ and H₃⁺ pathways, resulting from loss of one and two neutral hydrogen atoms.
- If all of these pathways are insignificantly small, then they should be mentioned, and the data should be provided as supplementary information.

Given that the dynamics being probed involve dynamics that likely occur at energies between 30-40 eV, did the authors consider a different electron impact energy?

- It seems 300 eV is excessive and the same should be observed following 70-300 eV energies, unless other states are involved. In that case, one wonders the relevance of this work to studies where the dication state is reached by laser excitation.

A great deal of the information provided about the production of H₃⁺ in this work derives from theory. This elicits the following questions:

- The authors should specify whether the DFT calculations used in the AIMD simulations were restricted or unrestricted. If restricted, the authors should discuss the fact that restricted DFT will disallow radical reactions.
- Furthermore, it is important how they prepared the initial state specifically. It seems trajectories

were initiated from the ground state of the cation, presumably following vertical excitation from the neutral ground state. If this is the case, the authors should justify why initiating a process near 35 eV (Fig. 2b) or 31.39 (Fig. 2c) is relevant following 300 eV electron impact. Why the discrepancy between Figs. 2b and 2c? Could the observed results come from this and many other potential energy surfaces?

- In Fig. 2, the blue and red energy levels are not explained in the caption.

- Finally, it would be instructive to have a statistical analysis of the trajectories in terms of percent H migration vs H₂ roaming.

The manuscript may be quite relevant to the understanding of high-energy chemical reactions and the prevalence of roaming. However, the authors should address the concerns raised here. The behavior of small hydrocarbons following high energy excitation via photons, ions, or electrons is evolving quickly. The authors should review if anything new has been published since submission.

Reviewer #3 (Remarks to the Author):

The authors describe a combined experimental and theoretical study of the ethane dication dynamics.

In the experimental part the authors provide information about the branching ratios of H_n⁺ + X⁺ products, as well as the KER distribution in the H₃⁺ formation.

In the theoretical part, the authors focus on the H₃⁺ formation dynamics and perform stationary TS point calculations as well as AIMD simulations. It is demonstrated that for ethane the roaming H₂ dynamics that result in H₃⁺ formation are accompanied by transfer of one of the H nuclei from one carbon to the other.

The simulated mechanism shown in the manuscript is of interest. Especially as it is significantly different from previously described mechanism in the MeOH dication, in which H₂ roaming is initiated from the CH₃ part and is in competition with proton transfer. Therefore the study provides new insight and in principle is publishable.

However, I have major concerns that should be addressed:

1. The AIMD simulations are carried out on relatively lower level single-reference calculation method, compared to previous works carried out using multi-reference approaches such as CASSCF and CASPT2 methods. Furthermore, viewing the “blue” curve (fig 2b) behavior at long distances it is clear that the calculation fails to describe the ground state which asymptotic behavior in the CH₄⁺ + CH₂⁺ limit should exhibit the long range Coulombic repulsion. This apparently erroneous behavior is most likely caused due to a multireference character of the actual ground state of the dissociated system, which can not be correctly described by the DFT based methods applied in this study. Thus a highly excited state with unbalanced charge distribution is reached by DFT.

2. Nevertheless, the KER in H₃⁺ formation is reasonably reproduced, indicating that it is quite possible that the important H₃⁺ forming dynamics are not very significantly affected and may provide some insight although performed using a single-reference method. To further support this

assumption it will be important to compare not only the KER in the H₃⁺ channel, but also the reported experimental relative rates of H₃⁺, H₂⁺ and H⁺ formation to the simulated branching ratios. Similarly, as it was strongly emphasized in the authors discussion, it will be important and valuable to report the ratio of C=C bond breaking vs H_n⁺ forming channels (and compare with exp if possible). Particular care should be taken, making sure of the fragment charge, as erroneous potentials can also result in a fictitious unbalanced charge distribution between the separated fragments.

3. In all the movies, the formation of a neutral H₂ seems to be made possible after (or in unison with) the migration of one of the hydrogens (or is it a proton ?) to form a CH₂-CH₄ structure. This point is new and interesting, as it is strikingly different from the roaming H₂ dynamics studied in methanol, in which proton migration and roaming H₂ mechanisms compete.

To convince that indeed (as proposed by the authors) the faster trajectories are going through the TS geometry, while the longer trajectories do not, it would be necessary to show some geometrical parameter as a function of time (e.g. the C=C angle rotation) that would clearly distinguish a TS mechanism from the others that go via a different path. I have to say that from the presented figures and movies I do not see a clear mechanistic distinction between short and long trajectories. And it is not convincing that the 100 fs is not an arbitrary choice, separating a single distribution into two parts. Did the authors observe a bimodal dissociation times distribution? Can these mechanisms be distinguished by their KER ?

4. Also in their introduction the authors create a contrast between two mechanisms – a transition state (TS) mechanism (e.g. reference 20) supported by KER measurements and a roaming mechanism supported by pump probe studies (e.g. refs 10-13). However, these two theoretical approaches describe practically identical dynamics:

The TS state calculated in ref 20 is actually on the path of the roaming mechanism, where a neutral H₂ is separated from the C₂H₄²⁺ dication. Thus, the roaming mechanism does proceed on the minimal energy path, in contrast to what is suggested by the authors. In fact, there's no difference between the mechanism proposed by ref 20 and the other references. The only difference is in the type of calculation performed to compare to the experimental data – an AIMD versus a TST modeling to obtain the KER.

Thus, suggesting that the literature shows two different pathways for forming H₃⁺ can mislead the readers and should be avoided.

5. Similarly, although this point is clearly seen in the roaming H movie, it would be much more convincing if the authors could present a quantitative measure that can clearly distinguish between the roaming H trajectories and the others. Thus proving that there's more than one distinct mechanism resulting in a broad dissociation time distribution.

Furthermore,

- in Fig 1: It's surprising that 300eV electrons do not result also in 3 body dissociation with H⁺ + C₂H_n⁺ with n=3 or 4. ? these are somehow missing from the coincidence map. This question relates also to the comparison of the relative probabilities of H₃⁺ / H₂⁺ and H⁺ formation with other studies e.g. ref 24 on line 97 (pg.3). As H₃⁺ formation can proceed on the ground state and

3-body fragmentation is more likely on the excited states, the H₃⁺ fraction in 2 body fragmentation can be expected to be higher than in all possible Coulomb explosion products that include also significant 3 body fragmentation where one of the fragments is an undetected neutral – see for example the different 2 and 3 body channels shown in ref 12 .

- On line 93, the end of the first paragraph of results, the authors write that the “coincidence stripes” are aligned with a -1 slope. This is must be a typo, since momentum conservation would dictate different recoil to the light and to the heavy fragment, resulting in a slope that depends on the fragment mass ratio. Inspection of fig 1 shows that indeed it is not a -1 slope. Maybe the authors intended to emphasize the anti-correlated negative slopes. Please correct the text.

- On line 105: The 4.7eV peak KER is stated to be larger than intense field studies – by how much ? is it significantly different ? what does it mean?

- In Fig 2b, the calculations made to produce the “red” and “blue” curves are not clearly defined. I assume that “red” means ground state with extension of the C-C bond from the FC geometry of the neutral CH₃-CH₃ geometry – this should be clearly explained. Were the other coordinates frozen ? or optimized ?

- The authors describe the dynamics on the “red” curve as having “high opportunity to dissociate via C-C bond 117 cleavage, due to the almost repulsive PES as a function of C-C bond length”. However, having zoomed in on the figure, it seems to me that once the doubly ionized system is formed at the FC geometry, there’s a significant few eV barrier for dissociation of the CC bond. In ref 12, a similar barrier is calculated for CH₃OH breakup and CC bond dissociation is attributed to dynamics on higher lying states. It seems that ethane might be similar.

- In contrast to the “red” curve, Fig2c shows a different initial energy at the neutral CC distance which is slightly lower than the “red” curve. I only guess it from the 31.39eV value in fig 2c, which is not discussed in the text. This is very confusing: Please make sure that all calculated results are explained to avoid confusing the readers. Looking at fig2c, it is now clear why the authors describe the CC bond dissociation as having “high opportunity” – although one still needs to explain the path, which is apparently not the direct dissociation presumably shown by the “red” curve and involves further specific structural rearrangement. (It is not clear from the text if this mechanism observed also in the AIMD ?)

- Fig 2c is most confusing as two very different reaction coordinates are superimposed. I strongly suggest separating the two paths – e.g. by showing one reaction coordinate in the positive X-axis direction and the other in the negative X-axis direction with respect to the initial FC geometry. Horizontal dashed lines can help to visually compare the minima and maxima of the two reaction pathways without superimposing them.

- It is not clear how the dissociation time is defined in the discussion of fig 3 – From the figure, it appears that some of the trajectories in figure 3a (assigned with <100fs H₃⁺ formation) actually seem to form H₃⁺ at longer times (some above 150fs). Please explain how and justify the way “dissociation time” is defined.

- Actually there should be two times that should be clearly identified by AIMD analysis – the separation time of the neutral H₂ and the proton capture which is followed by the rapid dissociation. Both appear as elongation of the mean distance of the three hydrogens that from the carbon, however, H₃⁺ formation can be clearly identified by looking at the fragments relative velocity which monotonically increases after H₃⁺ is formed due to the strong Coulombic repulsion. (Similarly there should be a way to characterize the times associated with the exciting

observation of roaming H dynamics)

- To continue my comment regarding the assignment of different mechanisms, it is not clear (at least from the supplied distance vs time figures 3a,b) why 100 fs was chosen – are there two clearly distinguishable dissociation time distributions ? It does not appear so from the 3a and 3b figures, making the chosen distinction seem arbitrary.
- It is not clear from figure 2c – is the transferred H is a proton or is it neutral ? Furthermore, what is the initial charge distribution at the FC geometry on the dication ground state ? is it balanced between the two CH₃⁺ parts ? or is it unbalanced as in the case of methanol ? The barrier shown by the “red” curve suggests that it might be unbalanced – similar to the methanol case where the barrier arises from the charge transfer – balancing it between the two fragments. In addition to the description of the geometrical changes during the AIMD simulation – one needs to provide more information about the charge distribution which is critical to the dynamics.
- Remarkably, as opposed to the proton migration previously reported for doubly ionized methanol – the authors provide evidence for neutral H migration (at least according to the “roaming neutral H” statement.) This should be emphasized and discussed.
- In the methods section – how many of the computed 1000 trajectories show H₃⁺ formation ?
- On line 224 – the TS model uses stationary point calculations; however the path is not stationary. Please correct the phrasing.

Response to referees

We would like to thank the referees for their careful reading, thoughtful suggestions, and thorough evaluation of our manuscript. We have addressed all comments and substantially revised the manuscript.

Reviewer #1 (Remarks to the Author):

This is a nice paper demonstrating roaming dynamics in the production of H₃⁺ from the dissociation of the ethane dication. I am please to recommend publication; however, I have a couple of comments that I would like the authors to consider.

Fig.2(a) shows very nice agreement between theory and AIMD simulations of the H₃⁺ KER distribution. Could they comment on differences, if any, in the KER for the roaming vs the IRC TS pathways. Similarly is the internal energy distribution of the H₃⁺ the same or different for these pathways. I do understand that the TS pathway is "minor" and so might make a small contribution to the KER, but deconstructing it according to pathway might be very instructive.

Response:

Thanks for the positive comments on our study. We have given the H₃⁺ KER distributions for different pathways (see section 3 in the supplementary information). The simulated KER of TS mechanism has the highest intensity at 4.5 eV which looks like to result in a sharp distribution, while the KER of the roaming process results in a flat distribution around 4.5 eV. Such a difference of KER might be observed employing an ultrafast pump-probe experiment.

"migrant" should be "migrate"

Response: We have corrected this mistake, thanks.

Reviewer #2 (Remarks to the Author):

The manuscript presents experimental and theoretical evidence regarding the formation of H₃⁺ from ethane following electron impact double ionization.

The experimental evidence is very clear, thanks to the isolation of two-body coincident dissociation events using COLTRIMS. The results in Fig. 1, are very clean. However, having removed all other coincidences, and not presenting a full mass spectrum, it would seem that 300 eV electron impact creates only three clean coincidences. All this information is missing.

- Of particular relevance to this study is the coincidence between CH₄⁺ and CH₂⁺ which would be suspected to be a major pathway for hydrogen migration.

- One also wonders about the C₂H₂⁺ and H₃⁺, and C₂H⁺ and H₃⁺ pathways, resulting from loss of one and two neutral hydrogen atoms.

- If all of these pathways are insignificantly small, then they should be mentioned, and the data should be provided as supplementary information.

Response: Figure 1 displays a small part of the full coincidence TOF map just for the targeted H_n⁺

($n=1-3$) cation channels, which result from two-body fragmentation of ethane dication. These three complete dissociation channels can be directly measured as no neutral fragment is involved. In addition, the branching ratios among these channels can be evaluated more accurately to make a comparison with those from other molecules (see Supplementary Table II).

We have presented the full coincidence TOF map in Supplementary Figure 1. According to this map, the number of counts for all ion-pair channels from the dissociation of ethane dication have been estimated in Supplementary Table I. From these counts, more information for the ethane dication dissociation can be obtained, e.g. the branching ratios among the H_n^+ ($n=1-3$) ions (see Supplementary Table II). Some of this information has been discussed in the main or supplemental texts.

- We have observed the $CH_2^+ + CH_4^+$ channel, which serves an indicator of hydrogen migration around the skeletal C-C bond as compared to the $CH_3^+ + CH_3^+$ channel. The count of the $CH_2^+ + CH_4^+$ channel is $\sim 6.3\%$ of that of the $CH_3^+ + CH_3^+$ channel and $\sim 16\%$ of that of the interested channel $H_3^+ + C_2H_3^+$. In the main text, we focus our attention on the $H_3^+ + C_2H_3^+$ channel.

- In addition to the dominant $H_3^+ + C_2H_3^+$ channel, other H_3^+ channels involving neutral hydrogen atom(s) are also observed. Their counts are also estimated as listed in Supplementary Table I. These many-body channels may exhibit more complicated dynamics which are beyond the scope of this article.

Given that the dynamics being probed involve dynamics that likely occur at energies between 30-40 eV, did the authors consider a different electron impact energy?

- It seems 300 eV is excessive and the same should be observed following 70-300 eV energies, unless other states are involved. In that case, one wonders the relevance of this work to studies where the dication state is reached by laser excitation.

Response:

We agree that the involved dynamics occurs at energies about 30-40 eV. The selected impact electron energy of 300 eV is more due to technical requirements. At this energy, the ionization cross-section is large enough and the pulsed electron beam is stable for a long time experiment. Reading from Ref. J. Chem. Phys. 109, 1704 (1998) (Table I), all of the possible channels are opened at impact energy higher than 45 eV and the partial ionization cross-section doesn't change so much as a function of impact energy.

In electron-molecule collisions, the deposition energy exhibits a broad distribution and so is the excitation energy of the doubly ionized molecule. Thus the dication could be produced in the ground state or many other excited states. This is usually dissimilar to the case of molecular photoionization where the excitation energy and thus precursor state can be selected. However, a previous photoionization study (see Ref. [14]) indicates that the H_3^+ channel of molecular dication is mainly caused by the ground state dissociation. For the case of electron impact (see Ref. [4] at an impact energy of 91 eV), the projectile energy loss spectrum shows onsets at about 28 eV for the H_3^+ channel from ethanol dication dissociation, which means that this channel is initiated by the removal of two electrons from the outermost molecular orbital reaching the dicationic ground state. Therefore, in the present study we simulate the dynamics of H_3^+ formation only on the potential energy surface of the ground state dication, the KER is well reproduced.

Generally speaking, the comparable parameters in electron collision and the photoionization are

the electron energy loss (energy difference between incident and scattering projectile) and the photon energy rather than the incident electron energy and the photon energy.

A great deal of the information provided about the production of H₃⁺ in this work derives from theory. This elicits the following questions:

- The authors should specify whether the DFT calculations used in the AIMD simulations were restricted or unrestricted. If restricted, the authors should discuss the fact that restricted DFT will disallow radical reactions.

Response: The DFT calculations used in the AIMD simulations were unrestricted. We have specified this in the theoretical method section.

- Furthermore, it is important how they prepared the initial state specifically. It seems trajectories were initiated from the ground state of the cation, presumably following vertical excitation from the neutral ground state. If this is the case, the authors should justify why initiating a process near 35 eV (Fig. 2b) or 31.39 (Fig. 2c) is relevant following 300 eV electron impact. Why the discrepancy between Figs. 2b and 2c? Could the observed results come from this an many other potential energy surfaces?

Response:

The simulations were performed for the dicationic ground state resulting from the vertical ionization of the neutral ground state of ethane. The initial configurations, velocities, and geometries of the atoms of the neutral ethane, are obtained by analyzing the normal mode vibration of the harmonic oscillator which is equivalent to the Wigner distribution.

As mentioned earlier, in electron-molecule collisions the dication could be produced in the ground state or many other states. Thus the observed results could come from other potential energy surfaces of ethane dication. Generally, based on the fast decay approximation the excitation energy can be regarded as the internal energy of the ground state molecule (see Ref. [6]) in the simulation to avoid an expensive calculation of the excited state. The fairly good agreement between the simulated and experimental KER of our present work indicates the initial condition adopted is reasonable.

The energy discrepancies between Fig. 2b and 2c are due to the following reasons: (1) Fig. 2b is the electronic energy without zero-point-energy (ZPE) correction while Fig. 2c includes the ZPE correction; (2) Fig. 2b is the rigid potential energy scan where all of the degrees of freedom are fixed while in Fig. 2c all of the stationary points (except for 31.39 eV) are obtained by geometry optimization. Strictly speaking, the energies in Fig. 2c are better than those in Fig. 2b. Fig. 2b is more a physical picture analysis showing that CH₃CH₃²⁺ is more likely to transform into CH₄CH₂²⁺ which is crucial for the production of H₃⁺.

- In Fig. 2, the blue and red energy levels are not explained in the caption.

Response: These energy levels have been explained in the caption.

- Finally, it would be instructive to have a statistical analysis of the trajectories in terms of percent

H migration vs H2 roaming.

Response: The H roaming is a very special mechanism and only one trajectory is confirmed among the 230 trajectories. In the revised manuscript we clarify that we only confirm one trajectory of this mechanism.

The manuscript may be quite relevant to the understanding of high-energy chemical reactions and the prevalence of roaming. However, the authors should address the concerns raised here. The behavior of small hydrocarbons following high energy excitation via photons, ions, or electrons is evolving quickly. The authors should review if anything new has been published since submission.

Response: We agree that the dynamics of small hydrocarbons by various excitation methods are a very hot topic. We have cited pertinent works in the revised manuscript.

Reviewer #3 (Remarks to the Author):

The authors describe a combined experimental and theoretical study of the ethane dication dynamics.

In the experimental part the authors provide information about the branching ratios of $H_n^+ + X^+$ products, as well as the KER distribution in the H_3^+ formation.

In the theoretical part, the authors focus on the H_3^+ formation dynamics and perform stationary TS point calculations as well as AIMD simulations. It is demonstrated that for ethane the roaming H_2 dynamics that result in H_3^+ formation are accompanied by transfer of one of the H nuclei from one carbon to the other.

The simulated mechanism shown in the manuscript is of interest. Especially as it is significantly different from previously described mechanism in the MeOH dication, in which H_2 roaming is initiated from the CH_3 part and is in competition with proton transfer. Therefore the study provides new insight and in principle is publishable.

Response: We thank the referee for the positive comments on our manuscript.

However, I have major concerns that should be addressed:

1. The AIMD simulations are carried out on relatively lower level single-reference calculation method, compared to previous works carried out using multi-reference approaches such as CASSCF and CASPT2 methods. Furthermore, viewing the “blue” curve (fig 2b) behavior at long distances it is clear that the calculation fails to describe the ground state which asymptotic behavior in the $CH_4^+ + CH_2^+$ limit should exhibit the long range Coulombic repulsion. This apparently erroneous behavior is most likely caused due to a multireference character of the actual ground state of the dissociated system, which cannot be correctly described by the DFT based methods applied in this study. Thus a highly excited state with unbalanced charge distribution is reached by DFT.

Response:

The CASSCF and CASPT2 works of Luzon *et al.* (Ref. [14]) and Livshits *et al.* (Ref. [15])

showed a fairly good description of fragmentation dynamics of methanol. However, due to the limited computational resources, it's hard for us to adopt the multi-reference method to simulate the dynamics for thousands of trajectories within a limited time. The propagation method like ADMP (atom-centered density matrix propagation which is similar to CPMD (Car–Parrinello molecular dynamics)) performed by DFT are reasonable choices for an order of thousand or even ten thousand trajectories. Such a DFT method at the same or similar level has been widely used to study the ultrafast hydrogen migration and the formation of H_3^+ , H_3O^+ ions from organic dications produced by electron impact (Ref. [4]), multiply charged ion collision (*J. Phys. Chem. Lett.* **4**, 3903 (2013)), and photoionization (Ref. [6]). The fairly good agreement between the theoretical and experimental KERs indicates that the calculation level of our present work is not bad.

For the potential energy curve of the dication, we analyze the charge distribution of $CH_2^+-CH_4^+$ channel as a function of bond length. The dissociation limit corresponds to $1+$ and $1+$ as shown by the following figures.

The potential energy starts to go down at C-C distance larger than 10 Å. In Fig. 2b, we perform the rigid potential energy scanning where all of the degrees of freedom are frozen. The obtained results might be different from the actual situation. In this figure, we want to show within or nearby the Franck-Condon region the hydrogen migration can stabilize the dication to form $[CH_2-CH_4]^{2+}$ which is crucial to forming the following H_3^+ .

For the dissociation of $[CH_2-CH_4]^{2+}$ into $CH_2^+ + CH_4^+$, a potential energy barrier which is close to the most probable vertical transition point is encountered. This channel is not likely to result from the dissociation of ethane dication in the ground state. This might explain that the experimentally observed $CH_2^+ + CH_4^+$ channel (see Supplementary Figure 1) is not accessed in the present AIMD simulations performed on the ground state potential energy surface.

2. Nevertheless, the KER in H_3^+ formation is reasonably reproduced, indicating that it is quite possible that the important H_3^+ forming dynamics are not very significantly affected and may provide some insight although performed using a single-reference method. To further support this assumption it will be important to compare not only the KER in the H_3^+ channel, but also the reported experimental relative rates of H_3^+ , H_2^+ and H^+ formation to the simulated branching ratios. Similarly, as it was strongly emphasized in the authors discussion, it will be important and valuable to report the ratio of C=C bond breaking vs H_n^+ forming channels (and compare with exp if possible). Particular care should be taken, making sure of the fragment charge, as

erroneous potentials can also result in a fictitious unbalanced charge distribution between the separated fragments.

Response:

First, we agree that we must pay particular care to analyze the final state of H_2 . Even the methods include the configuration interaction also provide a large divergence of the branching ratios of H_2 (see Table 1 in Re. [12]). The simulation results at 500 fs show that H_2 is produced in fractional charge (mostly 0.5 e). For the H_3^+ , it must be produced in 1+ state, (the neutral state of H_3 is not stable). For H^+ , the results show that it is produced in the 1+ state for ethane dication.

The counts or branching ratios of the C=C bond breaking and H_n^+ forming channels of ethane dication have been summarized in Supplementary Table I~III. Please keep in mind that in the present study we just simulate the dissociation dynamics of ethane dication in the ground state, which has been proven crucial to the formation of H_3^+ ions. In fact, in electron-molecule collisions the ethane dication could be populated in various states, which favor different dissociation channels. Therefore the present theoretical branching ratios extracted from the simulation of ground state do not agree well with the experimental ones.

Adopting the Mulliken population scheme, the fragment charge is obtained as the sum of the atomic charges in the corresponding fragment. Especially, we count all of the channels ending up with H_2 (might in neutral and charge states) which correspond to $H_2^+ + C_2H_4^+$ and $H_2 + C_2H_4^{2+}$ dissociation limit.

3. In all the movies, the formation of a neutral H2 seems to be made possible after (or in unison with) the migration of one of the hydrogens (or is it a proton ?) to form a CH2-CH4 structure. This point is new and interesting, as it is strikingly different from the roaming H2 dynamics studied in methanol, in which proton migration and roaming H2 mechanisms compete.

To convince that indeed (as proposed by the authors) the faster trajectories are going through the TS geometry, while the longer trajectories do not, it would be necessary to show some geometrical parameter as a function of time (e.g. the C=C angle rotation) that would clearly distinguish a TS mechanism from the others that go via a different path. I have to say that from the presented figures and movies I do not see a clear mechanistic distinction between short and long trajectories. And it is not convincing that the 100 fs is not an arbitrary choice, separating a single distribution into two parts. Did the authors observe a bimodal dissociation time distribution? Can these mechanisms be distinguished by their KER?

Response:

The single H migration results in $CH_2^+-CH_4^+$ structure, which is more stable than $CH_3^+-CH_3^+$ and the H migration is an ultrafast process. Therefore the geometry transformation is the most probable process due to its the largest energy gradient. The experimental coincidence channel of CH_2^+ and CH_4^+ , as shown in Supplementary Figure 1, confirms this possible geometry.

Generally speaking, if the H doesn't dissociate from its parent ion, it should be in a neutral state. Otherwise, there will be a Coulomb explosion due to the Coulomb repulsion. Therefore we want to ascribe this migration process as hydrogen migration.

There is no clear gap between TS and roaming mechanisms. Thus we agree that the choice of 100 fs is not a strict method. In the revised manuscript, we select the TS mechanism by

monitoring all of the H_3^+ trajectories. This is better than the single criterion of 100 fs. Of course, the method is qualitative rather than quantitative.

We don't observe a bimodal dissociation time distribution.

We analyze the KER distributions corresponding to TS and roaming mechanisms. As shown in Supplementary Figure 2, the KER of TS mechanism has the highest intensity at 4.5 eV which looks like to result in a sharp distribution, while the KER of the roaming process results in a flat distribution around 4.5 eV. We recommend a pump-probe experiment where there will be different KER distributions as a function of delay.

The manuscript is revised accordingly.

4. Also in their introduction the authors create a contrast between two mechanisms – a transition state (TS) mechanism (e.g. reference 20) supported by KER measurements and a roaming mechanism supported by pump probe studies (e.g. refs 10-13). However, these two theoretical approaches describe practically identical dynamics:

The TS state calculated in ref 20 is actually on the path of the roaming mechanism, where a neutral H_2 is separated from the $C_2H_4^{2+}$ dication. Thus, the roaming mechanism does proceed on the minimal energy path, in contrast to what is suggested by the authors. In fact, there's no difference between the mechanism proposed by ref 20 and the other references. The only difference is in the type of calculation performed to compare to the experimental data – an AIMD versus a TST modeling to obtain the KER.

Thus, suggesting that the literature shows two different pathways for forming H_3^+ can mislead the readers and should be avoided.

Response: We have revised the manuscript accordingly. The sentences to create a contrast between the two mechanisms are revised. On the other hand, we suggest that the dissociation times and KER distributions of these two mechanisms may be observed by a pump-probe experiment.

5. Similarly, although this point is clearly seen in the roaming H movie, it would be much more convincing if the authors could present a quantitative measure that can clearly distinguish between the roaming H trajectories and the others. Thus proving that there's more than one distinct mechanism resulting in a broad dissociation time distribution.

Response: We analyze all of the trajectories of H_3^+ channel and we only confirm one for the H roaming. It's not our purpose to separate the H roaming and H_2 roaming in this work because even we ascribe this trajectory to H roaming we also see the accompanying H_2 roaming as shown in Fig. 3d. A possible way to distinguish the H roaming is also expected by the pump-probe experiment where the large range of H roaming will result in low kinetic energy proton.

Furthermore,

• in Fig 1: It's surprising that 300eV electrons do not result also in 3 body dissociation with $H^+ + C_2H_n^+$ with $n=3$ or 4. ? these are somehow missing from the coincidence map. This question relates also to the comparison of the relative probabilities of H_3^+ / H_2^+ and H^+ formation with other studies e.g. ref 24 on line 97 (pg.3). As H_3^+ formation can proceed on the ground state and

3-body fragmentation is more likely on the excited states, the H_3^+ fraction in 2 body fragmentation can be expected to be higher than in all possible Coulomb explosion products that include also significant 3 body fragmentation where one of the fragments is an undetected neutral – see for example the different 2 and 3 body channels shown in ref 12 .

Response:

Figure 1 displays a small part of the full coincidence TOF map just for three two-body H_n^+ ($n=1-3$) cation channels. The full coincidence TOF map is presented in Supplementary Figure 1 where includes many three- or many-body dissociation channels. The number of counts for all dissociation channels has been estimated in Supplementary Table I. And thus the branching ratios among the H_n^+ ($n=1-3$) ions can be obtained (see Supplementary Table II), which has been discussed in the text.

In addition to the two-body channel $H_3^+ + C_2H_3^+$, other H_3^+ channels involving neutral hydrogen atom(s), e.g. $H_3^+ + C_2H_2^+/C_2H^+/CH^+$, are also observed. The fraction of $H_3^+ + C_2H_3^+$ is indeed much higher than those of other H_3^+ channels (see Supplementary Table I).

• On line 93, the end of the first paragraph of results, the authors write that the “coincidence stripes” are aligned with a -1 slope. This is must be a typo, since momentum conservation would dictate different recoil to the light and to the heavy fragment, resulting in a slope that depends on the fragment mass ratio. Inspection of fig 1 shows that indeed it is not a -1 slope. Maybe the authors intended to emphasize the anti-correlated negative slopes. Please correct the text.

Response: In our experiment, we only detect the ionic fragments from the molecule while the scattering projectile and ejected electrons are omitted. The final state electrons have only a small influence on the momentum conservation of the fragments. The magnitude of electron momentum is on the order of \sim a.u., while that of the fragment is in the order of several tens to hundreds of a.u. Please note that the vertical and horizontal scalars for TOF are different in Fig. 1. The slope of these three narrow coincidence stripes is indeed -1 (see the figure attached below for referee’s reference). The slope of the orange line is $(6.8 - 6.2)/(1.1 - 1.7) = -1$. The H_n^+ coincidence stripes aligned along the other three blue lines, which are parallel to the orange one, is indeed tilted with a slope of -1.

• On line 105: The 4.7eV peak KER is stated to be larger than intense field studies – by how much ? is it significantly different ? what does it mean?

Response: We have modified this sentence as “This peak value is larger than the previous results of about 4.0 and 4.3 eV obtained in strong laser field ionization [24,25]. This is probably caused by the drastic modification of the dication PES by the intense laser interaction, e.g., suppressing the dissociation barrier towards the H_3^+ fragments and thus making the original KER smaller.”

• In Fig 2b, the calculations made to produce the “red” and “blue” curves are not clearly defined. I assume that “red” means ground state with extension of the C-C bond from the FC geometry of the neutral CH_3-CH_3 geometry – this should be clearly explained. Were the other coordinates frozen ? or optimized ?

Response: We have defined the red and blue curves and explained them clearly in the figure caption. When extending the C-C internuclear distance, the other coordinates were frozen.

• The authors describe the dynamics on the “red” curve as having “high opportunity to dissociate via C-C bond 117 cleavage, due to the almost repulsive PES as a function of C-C bond length”. However, having zoomed in on the figure, it seems to me that once the doubly ionized system is formed at the FC geometry, there’s a significant few eV barrier for dissociation of the CC bond. In ref 12, a similar barrier is calculated for CH_3OH breakup and CC bond dissociation is attributed to dynamics on higher lying states. It seems that ethane might be similar.

• In contrast to the “red” curve, Fig2c shows a different initial energy at the neutral CC distance which is slightly lower than the “red” curve. I only guess it from the 31.39eV value in fig 2c, which is not discussed in the text. This is very confusing: Please make sure that all calculated results are explained to avoid confusing the readers. Looking at fig2c, it is now clear why the authors describe the CC bond dissociation as having “high opportunity” – although one still needs to explain the path, which is apparently not the direct dissociation presumably shown by the “red” curve and involves further specific structural rearrangement. (It is not clear from the text if this mechanism observed also in the AIMD ?)

Response:

Both of the above two comments are related to the Figs. 2b and 2c, so here we address them as a whole. Below is the reorganized discussion concerning the potential energy curves or diagrams as shown in Figs. 2b and 2c.

As shown by the reaction equations in Fig. 2b, the blue and red curves correspond to two reaction channels. This is a rigid potential energy scanning that freezes all of the degrees of freedom. Thus the energy may be different from the optimized one. This figure gives us a rough picture that, without hydrogen migration, the dication ion will dissociate into CH_3^+ and CH_3^+ along C-C bond stretching. The crossing between the red and blue curves indicates that hydrogen migration can transform the structure $[CH_3-CH_3]^{2+}$ to $[CH_2-CH_4]^{2+}$ where the latter is more stable and is the starting point of the H_3^+ emission channel as shown in Fig. 2c. The energy discrepancies

between Figs. 2b and 2c are due to the following reasons: (1) Fig. 2b is the electronic energy without zero-point-energy (ZPE) correction while Fig. 2c includes the ZPE correction; (2) Fig. 2b is the rigid potential energy scan where all of the degrees of freedom are fixed while in Fig. 2c all of the stationary points (except for 31.39 eV) are obtained by geometry optimization. Strictly speaking, the energies in Fig. 2c are better than those in Fig. 2b. Fig. 2b is more a physical picture analysis showing that $\text{CH}_3\text{CH}_3^{2+}$ is more likely to transform into $\text{CH}_4\text{CH}_2^{2+}$ which is crucial for the production of H_3^+ .

The associated texts in the manuscript have been revised to avoid misleading.

• Fig 2c is most confusing as two very different reaction coordinates are superimposed. I strongly suggest separating the two paths – e.g. by showing one reaction coordinate in the positive X-axis direction and the other in the negative X-axis direction with respect to the initial FC geometry. Horizontal dashed lines can help to visually compare the minima and maxima of the two reaction pathways without superimposing them.

Response: We have replaced Fig. 2c with a new version according to the referee's suggestion.

• It is not clear how the dissociation time is defined in the discussion of fig 3 – From the figure, it appears that some of the trajectories in figure 3a (assigned with <100fs H3+ formation) actually seem to form H3+ at longer times (some above 150fs). Please explain how and justify the way “dissociation time” is defined.

• Actually there should be two times that should be clearly identified by AIMD analysis – the separation time of the neutral H2 and the proton capture which is followed by the rapid dissociation. Both appear as elongation of the mean distance of the three hydrogens that from the carbon, however, H3+ formation can be clearly identified by looking at the fragments relative velocity which monotonically increases after H3+ is formed due to the strong Coulombic repulsion. (Similarly there should be a way to characterize the times associated with the exciting observation of roaming H dynamics)

• To continue my comment regarding the assignment of different mechanisms, it is not clear (at least from the supplied distance vs time figures 3a,b) why 100 fs was chosen – are there two clearly distinguishable dissociation time distributions ? It does not appear so from the 3a and 3b figures, making the chosen distinction seem arbitrary.

Response:

The above three comments point to the same concern, so here we address them as a whole.

We agree that the choice of 100 fs seems arbitrary. As mentioned before, in the revised manuscript we reanalyze all of the trajectories of H_3^+ channels, and the ones following (or very close to) the reaction coordinate of TS are ascribed to the TS mechanism. One more reasonable way to define the dissociation time is shown in Ref. Communications Chemistry 3, 49 (2020) where the oscillation-structure disappearing point on the relative velocity was defined as the dissociation time. Similarly, we define the dissociation time as the oscillation-structure disappearing point on the center of mass distance in the revised manuscript.

• *It is not clear from figure 2c – is the transferred H is a proton or is it neutral ? Furthermore, what is the initial charge distribution at the FC geometry on the dication ground state ? is it balanced between the two CH₃⁺ parts ? or is it unbalanced as in the case of methanol ? The barrier shown by the “red” curve suggests that it might be unbalanced – similar to the methanol case where the barrier arises from the charge transfer – balancing it between the two fragments. In addition to the description of the geometrical changes during the AIMD simulation – one needs to provide more information about the charge distribution which is critical to the dynamics.*

Response: The required charge states are shown in Fig. 2c. The charge distribution at the FC geometry on the dication ground state is balanced as a result of the symmetry of two CH₃⁺. But for the [CH₂-CH₄]²⁺, the CH₂ part carries a total charge of +0.85e and the CH₄ part carries +1.15e according to the Mulliken charge analysis at wb97xd/cc-pVTZ level. This means the transferred hydrogen atom leading to the isomerization from [CH₃-CH₃]²⁺ to [CH₂-CH₄]²⁺ carries a small fraction of one positive charge. We prefer to identify the transferred hydrogen as a neutral one.

• *Remarkably, as opposed to the proton migration previously reported for doubly ionized methanol – the authors provide evidence for neutral H migration (at least according to the “roaming neutral H” statement.) This should be emphasized and discussed.*

Response: We have modified corresponding sentences to discuss the H roaming process properly.

• *In the methods section – how many of the computed 1000 trajectories show H₃⁺ formation ?*

Response: We have added a sentence in the theoretical method section “A total of 1000 trajectories was computed and 230 of them end up with H₃⁺ formation.”.

• *On line 224 – the TS model uses stationary point calculations; however the path is not stationary. Please correct the phrasing.*

Response: We have modified this sentence.

Reviewers' comments:

Reviewer #1 (Remarks to the Author):

The authors have done a thorough revision and I believe have addressed all concerns. Clearly the roaming pathway to H₃⁺ is not a major one at the energy of the experiment; however, it is nicely supported by the trajectory simulations. I would also note that in general lifetimes are not definitive signatures of "roaming", which in fact can be quite prompt.

I do recommend publication.

Reviewer #2 (Remarks to the Author):

The authors have addressed each of the questions in detail to the best of their available information. There may still be questions that remain unanswered and will need new experiments and new calculations, but those are not needed to accept the paper for publication.

Reviewer #3 (Remarks to the Author):

The authors clearly made an effort to improve their manuscript.

Nevertheless, in my opinion, at least some of the details included in response to the review, should be addressed also in the manuscript itself.

In particular, I think that the charge distribution presented in the authors response letter should be included at least in the supporting information.

I find it highly unlikely and surprising that two cations (with frozen coordinates) attract each other at 3-6Å distances - as shown by the blue curve in figure 2b. I would expect the potential to be dominated by the long range Coulombic repulsion at this long distance. Nevertheless, the authors claim in their response that the charge is almost equally distributed between a CH₂⁺ and CH₄⁺ fragments. I suggest that the authors should at least point out this surprising behaviour to the readers in the manuscript text itself and add the charge distribution they calculated to support their assignment. (to be clear, I'm attaching a copy of that figure)

One more point I would like to contribute is the fact that a system is symmetric does not necessarily result in equal charge distribution. The symmetric CH₃-CH₃ system can easily find itself in a superposition of non-symmetric states: i.e. dication-neutral + neutral-dication. Furthermore, considering the role of symmetry, it might be instructive to perform static PES calculations not only in the symmetric minimum energy geometry , but also from other regions in the FC region in which symmetry can be broken by the zero point motion.

Response to referees

We would like to thank the referees for their careful reading, thoughtful suggestions, and thorough evaluation of our revised manuscript. We have addressed all comments and revised the manuscript again. The revised sentences are highlighted in blue color.

Reviewer #1 (Remarks to the Author):

The authors have done a thorough revision and I believe have addressed all concerns. Clearly the roaming pathway to H_3^+ is not a major one at the energy of the experiment; however, it is nicely supported by the trajectory simulations. I would also note that in general lifetimes are not definitive signatures of "roaming", which in fact can be quite prompt.

I do recommend publication.

Response:

We thank the reviewer for the positive evaluation of our manuscript. We agree that, in general, lifetimes are not definitive signatures of "roaming", which in fact can be quite prompt. As shown by the trajectories of our simulation, a small part of the roaming mechanism emit the H_3^+ near 100 fs which is an even fast process. In the revised manuscript, we modify some sentences which are highlighted in blue color (line 28, and line 231).

Reviewer #2 (Remarks to the Author):

The authors have addressed each of the questions in detail to the best of their available information. There may still be questions that remain unanswered and will need new experiments and new calculations, but those are not needed to accept the paper for publication.

Response:

We thank the reviewer for the positive evaluation of our manuscript.

Reviewer #3 (Remarks to the Author):

The authors clearly made an effort to improve their manuscript.

Nevertheless, in my opinion, at least some of the details included in response to the review, should be addressed also in the manuscript itself.

In particular, I think that the charge distribution presented in the authors response letter should be included at least in the supporting information.

I find it highly unlikely and surprising that two cations (with frozen coordinates) attract each other at 3-6Å distances - as shown by the blue curve in figure 2b. I would expect the potential to be dominated by the long range Coulombic repulsion at this long distance. Nevertheless, the

authors claim in their response that the charge is almost equally distributed between a CH_2^+ and CH_4^+ fragments. I suggest that the authors should at least point out this surprising behaviour to the readers in the manuscript text itself and add the charge distribution they calculated to support their assignment. (to be clear, I'm attaching a copy of that figure) One more point I would like to contribute is the fact that a system is symmetric does not necessarily result in equal charge distribution. The symmetric $\text{CH}_3\text{-CH}_3$ system can easily find itself in a superposition of non-symmetric states: i.e. dication-neutral + neutral-dication. Furthermore, considering the role of symmetry, it might be instructive to perform static PES calculations not only in the symmetric minimum energy geometry, but also from other regions in the FC region in which symmetry can be broken by the zero point motion.

Response:

We thank the reviewer for the informative comments and suggestions.

1. We revise Fig. 2 in the manuscript to include the relaxed potential energy scanning and the charge distribution as a function of bond length. The corresponding manuscript is revised (line 130-140 and line 150-162).

2. For the no Coulombic repulsion behavior at a large distance, first, we want to exclude the influence of the basis set. We perform the calculations using $\omega\text{B97XD/aug-cc-pVTZ}$ which includes the diffuse basis set. Generally, the missing diffuse basis set may result in an inaccurate description of the potential energy at a very large distance. The new results show similar behavior as that of $\omega\text{B97XD/cc-pVTZ}$. Here, we ascribe the no Coulombic repulsion behavior due to the frozen of all of the degrees of freedom. In this case, the geometries of each moiety and their relative orientation are fixed which might result in a large dipole attractive force between them. We analyze the charge distribution of this CH_4^+ . At a large distance (e.g. 3-6 Å), carbon is in a negative charge state which will attract the facing CH_2^+ moiety. We perform a relaxed PEC scan in the revised manuscript. The new result shows that the Coulombic repulsion behavior is obtained at distance larger than 4.6 Å. In the revised manuscript, the data are updated to the results of $\omega\text{B97XD/aug-cc-pVTZ}$.

3. To break the symmetry, we perform a relaxed PES scan without symmetry restriction of the $\text{CH}_3^+ + \text{CH}_3^+$ channel. For each of the scan steps, the symmetry is C_1 and equal charge separation is also obtained. Furthermore, to include the molecular vibration during the dissociation we simulate one trajectory of $\text{CH}_3^+ + \text{CH}_3^+$ channel using ADMP (B3LYP/cc-pVDZ) method. The molecular geometries of each step are extracted and used to perform Mulliken charge analysis by $\omega\text{B97XD/aug-cc-pVTZ}$. The results also show an almost equal charge separation for the dissociation limit. The expression of the relationship between charge and symmetry is revised. For the ground state of the dication of ethane $[\text{CH}_3\text{-CH}_3]^{2+}$, the density functional method together with the aug-cc-pVTZ basis provides a good description of the wavefunction and thereby the charge state. We believe the “dication-neutral + neutral-dication” configuration will result in no-equal charge separation. Such a configuration, however, is most likely corresponding to an excited state which is out of our current discussion.

Reviewers' comments:

Reviewer #3 (Remarks to the Author):

As requested, the authors discussed the unexpected shape of the PES curve, making important improvements to the figures and text of the manuscript.

It is now suitable for publication as the limitations of the theoretical treatment are clearly emphasized.

Response to referees

Reviewer #3 (Remarks to the Author):

As requested, the authors discussed the unexpected shape of the PES curve, making important improvements to the figures and text of the manuscript.

It is now suitable for publication as the limitations of the theoretical treatment are clearly emphasized.

Response:

We thank the reviewer for the positive evaluation of our manuscript.